# Qki activates Srebp2-mediated cholesterol biosynthesis for maintenance of eye lens transparency

Seula Shin[1,2], Hao Zhou [3], Chenxi He[4], Yanjun Wei [5], Yunfei Wang[6], Takashi Shingu[1], Ailiang Zeng[1], Shaobo Wang[7], Xin Zhou[8], Hongtao Li[9], Qiang Zhang[1], Qinling Mo[3], Jiafu Long [3], Fei Lan[4], Yiwen Chen [5] & Jian Hu [1,2,10] ✉

Defective cholesterol biosynthesis in eye lens cells is often associated with cataracts; however, how genes involved in cholesterol biosynthesis are regulated in lens cells remains unclear. Here, we show that Quaking (Qki) is required for the transcriptional activation of genes involved in cholesterol biosynthesis in the eye lens. At the transcriptome level, lens-specific Qki-deficient mice present downregulation of genes associated with the cholesterol biosynthesis pathway, resulting in a significant reduction of total cholesterol level in the eye lens. Mice with Qki depletion in lens epithelium display progressive accumulation of protein aggregates, eventually leading to cataracts. Notably, these defects are attenuated by topical sterol administration. Mechanistically, we demonstrate that Qki enhances cholesterol biosynthesis by recruiting Srebp2 and Pol II in the promoter regions of cholesterol biosynthesis genes. Supporting its function as a transcription co-activator, we show that Qki directly interacts with single-stranded DNA. In conclusion, we propose that Qki-Srebp2–mediated cholesterol biosynthesis is essential for maintaining the cholesterol level that protects lens from cataract development.

[1] Department of Cancer Biology, The University of Texas MD Anderson Cancer Center, Houston, TX, USA. [2] Cancer Biology Program, MD Anderson Cancer Center UTHealth Graduate School of Biomedical Sciences, Houston, TX, USA. [3] State Key Laboratory of Medicinal Chemical Biology, Tianjin Key Laboratory of Protein Science, College of Life Sciences, Nankai University, Tianjin, China. [4] Shanghai Key Laboratory of Medical Epigenetics, International Co-laboratory of Medical Epigenetics and Metabolism, Ministry of Science and Technology, Institutes of Biomedical Sciences, Fudan University, and Key Laboratory of Carcinogenesis and Cancer Invasion, Ministry of Education, Liver Cancer Institute, Zhongshan Hospital, Fudan University, Shanghai, China. [5] Department of Bioinformatics and Computational Biology, The University of Texas MD Anderson Cancer Center, Houston, TX, USA. [6] Clinical Science Division, H. Lee Moffitt Cancer Center & Research Institute, Tampa, FL, USA. [7] Department of Neurosurgery, Qilu Hospital, Cheeloo College of Medicine, Shandong University, Jinan, Shandong, China. [8] Cancer Research Institute of Jilin University, The First Hospital of Jilin University, Jilin, China. [9] Department of Oncology, Affiliated Sixth People's Hospital, Shanghai Jiaotong University, Shanghai, China. [10] Neuroscience Program, MD Anderson Cancer Center UTHealth Graduate School of Biomedical Sciences, Houston, TX, USA. ✉email: jhu3@mdanderson.org

Cholesterol is an essential building block of mammalian cell membranes and plays an important role in multiple cellular functions, such as membrane raft signaling transduction[1], intracellular vesicle trafficking[2], and cell growth[3]. Cells obtain cholesterol through cholesterol biosynthesis and cholesterol uptake[2,4]. Whereas most tissues use both of these mechanisms to meet the needs for cellular cholesterol[2], some cholesterol-rich tissues, such as eye lens and brain, in which the supply of cholesterol from plasma lipoproteins is stringently limited, depend extensively on de novo cholesterol biosynthesis[5,6]. However, how these tissues meet the high demand for cholesterol biosynthesis remains largely unknown.

Sterol regulatory element-binding protein 2 (SREBP2) is the major transcription factor that regulates cholesterol biosynthesis[7]. A number of proteins are suggested to regulate the processing and transcriptional activity of SREBP2 in the liver, such as SREBP cleavage-activating protein (SCAP), progestin and adipoQ receptors 3 (PAQR3), CREB-binding protein (CBP), hepatocyte nuclear factor-4 (HNF-4), and small heterodimer partner (SHP)[8–12]. Although SREBP2 is ubiquitously expressed, SREBP2-mediated cholesterol biosynthesis seems to depend on different regulators in different tissues. For example, glycerol kinase 5 (GK5), a skin-specific kinase, regulates SREBP2 processing and controls cholesterol homeostasis in sebocytes, terminally differentiated epithelial cells in sebaceous glands that are required for normal hair follicle differentiation and cycling[13]. Hence, different tissues are impacted by dysregulated cholesterol biosynthesis to different extents. For instance, hypercholesterolemia often leads to cardiovascular diseases in humans[14], whereas insufficient cholesterol levels caused by various hereditary mutations of cholesterol biosynthesis genes preferentially lead to cataracts in the eye lens[5]. Consistently, lipid composition analysis revealed that eye lens cells contain a larger portion of cholesterol than do other cell types, strongly implicating a high demand for cholesterol in the lens[15]. Therefore, studying cholesterol biosynthesis in the eye lens may identify transcriptional regulators of SREBP2 and provide insight into tissue-specific cholesterol homeostasis.

Maintaining the transparency of eye lens is essential to properly reflect light onto the retina. Crystallins are the most abundant and highly water-soluble lens proteins, which form a transparent lens structure. Particularly, α-crystallins function as molecular chaperons that facilitate proper folding of β- and γ-crystallins[16]. Aberrant folding of the crystallins leads to pathogenic protein aggregates, eventually manifesting as cataracts[17]. Therefore, studying how to enhance the function of chaperones, mainly α-crystallins in eye lens can be an effective strategy to develop noninvasive treatment for cataracts. Notably, recent studies demonstrated that lanosterol and other sterol derivatives could reduce crystallin aggregates by stabilizing the α-crystallins[18,19]. Importance of cholesterol biosynthesis in prevention of cataractogenesis is also implicated in various human syndromes and pharmacological treatments[5,20,21]. Various genetic syndromes that manifest as cataracts are caused by mutations of cholesterol biosynthesis genes[5,22,23]. For instance, it has been reported that missense mutations of lanosterol synthase (LSS), a cyclization enzyme in cholesterol biosynthesis, caused cataracts in a pedigree analysis of consanguineous family of Caucasian descent[19]. Also, about 20% of patients with Smith-Lemli-Opitz syndrome, who are deficient in 7-dehydrocholesterol reductase (DHCR7), have cataracts[24,25]. It has also been reported that 30% of patients with mevalonic aciduria due to mevalonate kinase (MVK) deficiency presented with cataracts[26]. Additionally, lathosterolosis and microcephaly, congenital cataract, and psoriasiform dermatitis (MCCPD), caused by deficiency in either sterol-C5-desaturase (SC5D) or methylsterol monoxygenase 1 (MSMO1), is known to display cataracts as one of the clinical features[27,28]. Lastly, 33.7%

of patients who received treatment with statins, inhibitors of 3-hydroxy-3-methylglutaryl-CoA reductase (HMGCR) that suppress cholesterol biosynthesis, have cataracts as a side effect[29,30]. Similar to human patients, multiple animal models with defects in cholesterol biosynthesis genes such as Srebf2, Lss, and squalene synthase (Fdft1) have displayed cataractogenic phenotypes[31,32]. Despite genetic and pharmacological studies supporting the essential role of cholesterol biosynthesis in regulating crystallin folding and lens transparency, understanding the molecular mechanisms underpinning the transcriptional regulation of cholesterol biosynthesis genes in the eye lens is rather limited.

In addition to the genetic diseases attributable to mutations of genes known to be involved in cholesterol biosynthesis, 6q deletion syndrome displays cataracts as one of the clinical features[33,34]. Although multiple genes reside in the 6q terminal deletions, breakpoint of a single gene, Quaking (QKI), has been shown to cause a clinical phenotype highly similar to the common 6q deletion syndrome phenotypes, suggesting that QKI loss plays a pathogenic role in 6q deletion syndrome[35]. QKI is a member of the signal transduction and activation of RNA (STAR) family of proteins and is involved in mRNA metabolism in a tissue-specific manner including regulation of myelination in central nervous system, adipose tissue metabolism, monocyte differentiation, and endothelial barrier function[36–40]. Our previous studies also demonstrated that QKI functions as a tumor suppressor by regulating the endolysosomal pathway in neural stem cells (NSCs) and glioma stem cells[41]. Because deletion of QKI is linked with 6q deletion syndrome in humans, in the present study, we sought to determine whether and how QKI loss causes cataractogenesis.

Here, we show that Qk deficiency in the lens epithelium leads to cataracts with 100% penetrance. The cataracts formed in Qk-deficient mice are caused by a reduced cholesterol level, which can be rescued by administering sterol eye drops to the mice. Mechanistically, we have identified Qki as a transcriptional regulator of Srebp2-mediated transcription of cholesterol biosynthesis genes. In addition, we show that Qki directly interacts with DNA to regulate transcription.

## Results

**Deletion of Qk in eye lens cells leads to cataracts.** To investigate the mechanism underpinning Qki's regulation of protein homeostasis in vivo, we chose the Nestin-CreER$^{T2}$; Qk$^{L/L}$ mouse model (hereafter denoted as "Qk-iCKO")[41], as Nestin is a specific marker of ectodermal lineage, that of lens epithelial cells (LECs) and NSCs (Supplementary Fig. 1a)[42–44]. We confirmed that LECs express ectodermal markers such as Nestin and GFAP using immunofluorescent staining (Supplementary Fig. 1b)[45,46]. To determine whether this model has deletion of Qk in lens cell lineages specifically and efficiently, we crossed Qk-iCKO mice with the Rosa26-loxP-mTRed-STOP-loxP-mGFP reporter line[41,47], which enabled us to trace Qki-depleted cells according to the expression of membrane-bound green fluorescent protein (GFP). First, we found that one of the Qki isoforms, Qki-5 is specifically expressed in the lens cells and predominantly localized to the nucleus (Fig. 1a). In the Qk-iCKO mice injected with tamoxifen (tam) at postnatal day 7 (P7), Qki-5 expression was nearly 100% depleted in all GFP$^+$ lens cells at P19 (Fig. 1a, b and Supplementary Fig. 1c, d). Of note, another isoform, Qki-6 was highly localized to the nucleus similar to Qki-5, but the expression of Qki-6 was only 30% decreased in GFP$^+$ lens cells of Qk-iCKO mice at P19 (Supplementary Fig. 1e, f), potentially due to the relatively higher stability of Qki-6 compared to Qki-5, suggesting that depletion of Qki-5 is responsible for the phenotypic alteration in eye lens from the Qk-iCKO mice shown below.

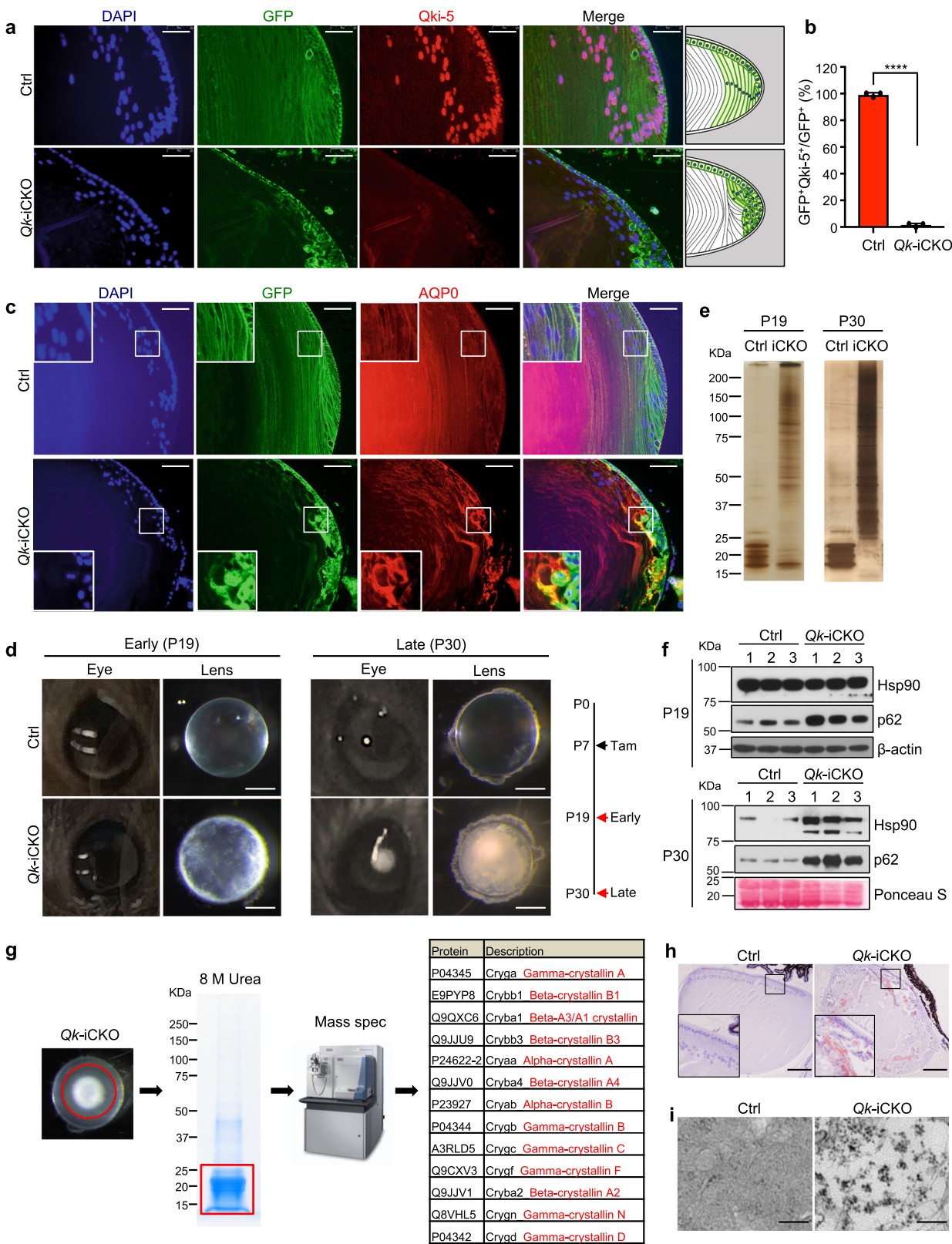

Additionally, we demonstrated that the membrane-bound GFP is localized to the membrane shown by co-staining with the lens-specific membrane protein called aquaporin 0 (AQP0)[48] in control mice, which is consistent with the previous study reporting the mTmG reporter allele[47] (Fig. 1c and Supplementary Fig. 2a). Lens membrane structure was largely affected upon Qki depletion in lens fiber cells (LFCs) with mislocalization of membranes,

indicated by the staining of both GFP and AQP0 (Fig. 1c and Supplementary Fig. 2a). Morphologically, the LFCs with Qki depletion failed to elongate and migrate toward the center of the body and showed aberrant directionality (Fig. 1a). As a regularly organized membrane structure in the lens tissue is essential for the lens transparency[49], we reasoned that the impairment of lens membrane integrity with Qki depletion might lead to crystallin

**Fig. 1 Deletion of *Qk* in eye lens cells leads to cataracts. a** Representative immunofluorescent stains of lens cells in paraffin-embedded sections of ocular tissue from control (Ctrl) and *Qk*-iCKO mice for GFP (green) and Qki-5 (red) (scale bar: 50 μm) at P19. DAPI (blue): nuclei. Schematic at the far right depicting the morphological impairment of eye lens structure upon Qki loss in *Qk*-iCKO mice compared to the control mice. **b** Quantification of GFP+Qki5+ cells among all GFP+ cells in the eye lens tissue represented in **a**. *n* = 3 mice/group. The results are presented as means with standard deviation (SD). *p* = 0.0000001090. ****p < 0.0001 (two-tailed unpaired t-test). **c** Representative immunofluorescent stains of lens cells in paraffin-embedded sections of ocular tissue from control and *Qk*-iCKO mice for GFP (green) and AQP0 (red) (scale bar: 50 μm) at P19. DAPI (blue): nuclei. **d** Representative images of eyes and eye lenses isolated from Ctrl and *Qk*-iCKO mice at early (P19) and late (P30) timepoints. Scale bar: 0.5 mm. **e** RIPA-soluble fractions of protein lysates from the isolated lenses of Ctrl and *Qk*-iCKO mice at early (P19) and late (P30) timepoints visualized using silver staining. **f** Immunoblots of lenses isolated from Ctrl (*n* = 3) and *Qk*-iCKO (*n* = 3) mice at P19 and P30 for detection of the proteostatic stress markers heat shock protein 90 (Hsp90) and p62. β-actin and Ponceau S: loading control. **g** Aggregates extracted from *Qk*-iCKO lenses (red circle) dissolved in 8 M urea and visualized using Coomassie blue staining. The strong Coomassie blue-stained band (red square) was excised and subjected to mass spectrometric (MS) analysis. The most enriched proteins according to MS analysis were ranked as shown in the table. The image of MS machine (Orbitrap Pro) is from Thermo Fisher Scientific website. **h, i** Representative images of **h** Congo Red-stained (scale bar: 200 μm) and **i** transmission electron microscopy-analyzed (scale bar: 250 nm) Ctrl and *Qk*-iCKO lenses at P30. All the experiments were replicated three times in the lab.

aggregation, and ultimately cause cataract phenotype in the *Qk*-iCKO mice. Indeed, all *Qk*-iCKO mice receiving P7 tam injections exhibited cloudy eye lenses starting at P19, which progressed to massive aggregates in the lenses at P30 while the control mice (including *Nestin-CreER^T2^; Qk^+/+^, Nestin-CreER^T2^; Qk^L/+^*, and *Qk^L/L^*) displayed transparent and healthy lenses (Fig. 1d).

Because cataracts are known to be initiated from soluble, but covalent multimers of crystallins in human eye lens[50,51], we first asked if the lens proteins display soluble multimeric conformation upon *Qk* deletion. We showed that loss of *Qk* in lens cells led to significant increase in higher molecular weights in RIPA buffer-soluble fractions of total proteins compared to controls in a time-dependent manner, suggesting that the initial protein aggregates upon *Qk* deletion are soluble in RIPA buffer (Fig. 1e). Furthermore, immunoblotting showed that *Qk*-deficient lens cells were progressively enriched in proteostatic stress markers such as heat shock protein 90 (hsp90), p62, and ubiquitin (Fig. 1f and Supplementary Fig. 3a, b)[52]. Ultimately, we found that Qki depletion-induced protein aggregates in the lenses at the later timepoint (after 1 month) were RIPA buffer-insoluble and only dissolved in 8 M urea (Fig. 1g) as suggested in human cataract lens[51]. To further test if the protein aggregates in *Qk*-deficient lenses recapitulate crystallin aggregates in human cataract lens[17], we dissolved the aggregates isolated from the Qki-depleted lens tissue of the later timepoint using a strong denaturant (8 M urea) (Fig. 1g). Mass spectrometric analysis demonstrated that the vast majority of the urea-denatured protein aggregates in *Qk*-deficient lenses belonged to the family of crystallins, including various isoforms of α-, β-, and γ-crystallins (Fig. 1g), which are the major components of human and mouse cataracts, suggesting that these aggregates are bona fide cataracts in *Qk*-deficient lenses. We also confirmed that mRNA and protein levels of crystallin isoforms were not significantly changed upon Qki depletion (Supplementary Fig. 3c–e), suggesting that it is not the alteration of crystallin expression per se that caused abnormal protein aggregation. We further characterized the protein aggregates induced by *Qk* deletion as forms of amyloid fibrils using Congo Red staining (Fig. 1h); this is the most common form of crystallin aggregation, resulting in cataracts[53]. In addition, transmission electron microscopy confirmed that the protein aggregates in *Qk*-deficient lenses displayed amyloid-like fibrils (Fig. 1i)[18]. Taken together, these data suggested that Qki is required for maintaining proper protein folding in lens cells, which is essential for the transparency of the lens.

**Qki is a transcriptional regulator of the cholesterol biosynthesis pathway in eye lens.** To identify the mechanisms by which Qki regulates protein homeostasis in lens cells, we performed transcriptomic profiling of lens cells isolated from control and *Qk*-iCKO mice. We selected the timepoint (P17-19) when cataract phenotype is apparent in *Qk*-deficient lens cells and secondary effects of Qki loss on the transcriptome can be minimized. Cholesterol biosynthesis was the most enriched downregulated pathway upon *Qk* deletion as determined using Ingenuity Pathway Analysis (IPA) (Fig. 2a). Importantly, lens cell membrane has a molar ratio of cholesterol/phospholipids that is 2–4-fold higher than those in the other typical cell membranes[15], and the unusually abundant cholesterol in lens membrane is critical for α-crystallin binding to the lens membrane for its chaperone activity to prevent from abnormal crystallin aggregation[54]. Therefore, we decided to focus on cholesterol biosynthesis pathway as a potential target of Qki, which is impaired in the lenses from *Qk*-iCKO mice, which might lead to cataract phenotype. Notably, transcriptomic profiling data demonstrated that 14 of total 19 enzymes involved in the steps in cholesterol biosynthesis decreased at the mRNA level upon Qki loss (Fig. 2b, c). We confirmed that Qki loss significantly decreased the expression of 14 genes involved in cholesterol biosynthesis as demonstrated by reverse transcription-quantitative polymerase chain reaction (RT-qPCR) (Fig. 2d). Consistently, protein expression levels of cholesterol biosynthesis enzymes, such as hydroxymethylglutaryl-CoA synthase (Hmgcs1), Hmgcr, and farnesyl diphosphate synthase (Fdps), were also greatly reduced (Fig. 2e, f). Furthermore, immunofluorescent staining demonstrated dramatically decreased Hmgcs1 expression in lens cells upon Qki depletion (Supplementary Fig. 3f–i). In accord with the in vivo findings in our mouse model, downregulation of the cholesterol biosynthesis pathway was also evident in human lens epithelial cell line (HLE-B3) cells in which *QKI* was deleted using the CRISPR-Cas9 system (Supplementary Fig. 4a–c). RNA sequencing (RNA-seq) also demonstrated reduced expressions of cholesterol biosynthesis genes in *QKI*-deleted HLE-B3 cells. Because HLE-B3 is an immortalized cell line, we utilized the NSC-derived lens progenitor-like cells (NLPCs) from *Qk*-iCKO mice to better recapitulate our in vivo finding for the further molecular studies. Pax6^high^ NSCs differentiated into Pax6^intermediate^ αB-crystallin^high^ NLPCs when we induced with stepwise growth factor-based treatment (first step: Noggin; second step: BMP4, BMP7, and bFGF) for induction of lens cell lineage (Fig. 2g)[55]. Immunofluorescent staining demonstrated that Qki expression was completely abolished in NSCs and NLPCs by treatment with 4-hydroxytamoxifen (Fig. 2h, Supplementary Fig. 4d). Consistent with the observation in mouse eye lens tissue and HLE-B3 cells, cholesterol biosynthesis was also one of the most downregulated pathways in NLPCs upon *Qk* deletion (Fig. 2i). The majority of the cholesterol biosynthesis genes (16 of 19) were downregulated at the mRNA level in Qki-depleted NLPCs compared to wild-type

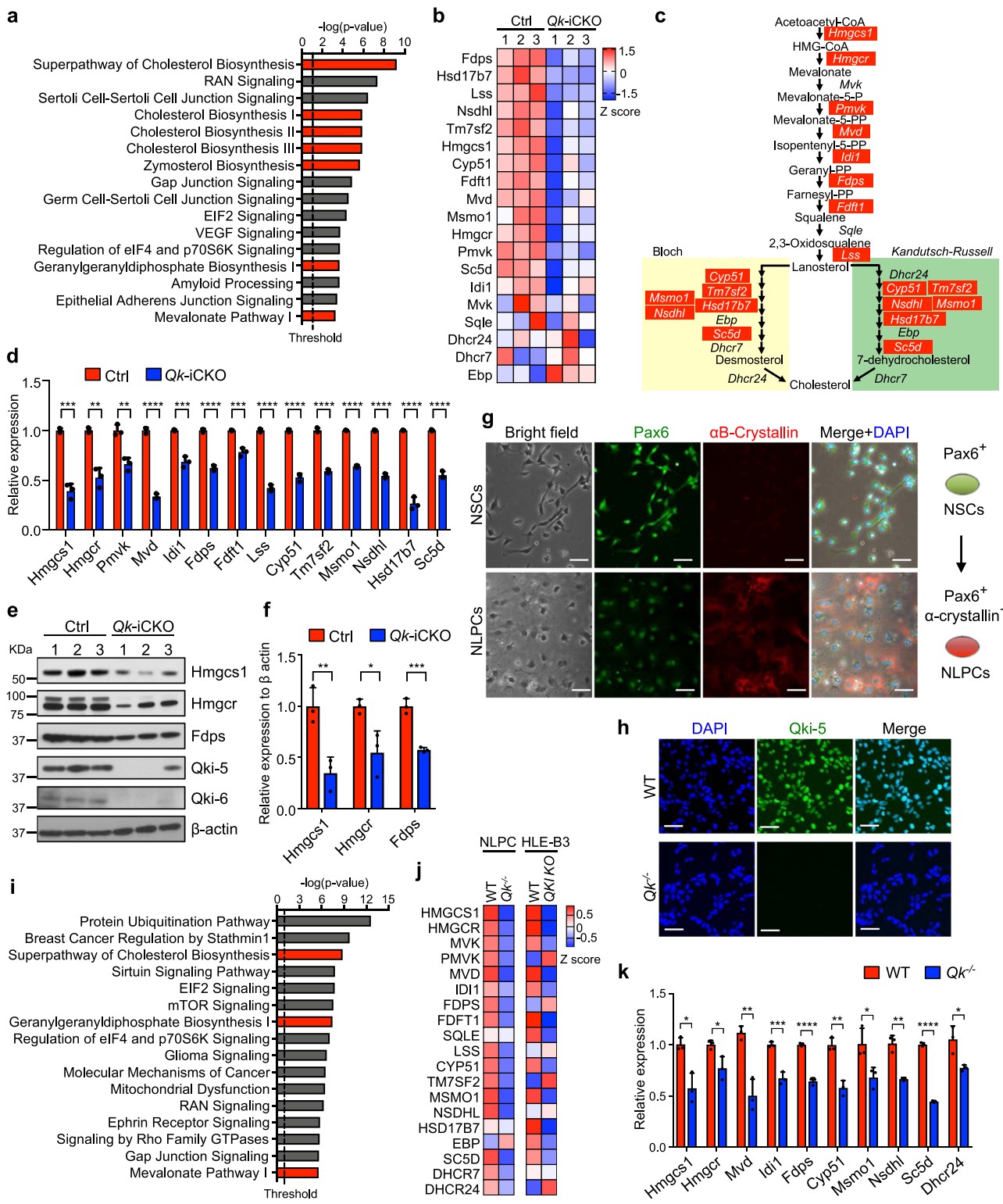

(WT), which was in line with the RNA-seq data in HLE-B3 cells (Fig. 2j). Consistently, RT-qPCR validated the reduced mRNA expression of cholesterol biosynthesis genes enriched in RNA-seq in NLPCs (Fig. 2k).

We next asked whether downregulation of cholesterol biosynthesis genes induced by *Qk* deficiency results in a decreased cholesterol level. We found that the cellular cholesterol level as visualized using filipin staining was reduced upon *Qk* deletion from lens cells at P19 (Fig. 3a, b). Consistently, the total free

cholesterol level was markedly lower in lenses isolated from *Qk*-iCKO mice than in those from the control mice at P19 (Fig. 3c). Of note, we did not observe the systemic alteration of cholesterol level in brain and plasma at P19 (Supplementary Fig. 4e, f), suggesting the lens-specific role of Qki in regulating cholesterol level. Recent studies have reported the potential role of sterol molecules in alleviating cataract formation[18,19]. To determine whether replenishment of sterols leads to alleviation of cataract formation induced by *Qk* deletion, we applied lanosterol drops to

**Fig. 2 Qki depletion leads to downregulation of genes in the cholesterol biosynthesis pathway in eye lens. a** IPA of the genes with a significant reduction of mRNA expression in isolated *Qk*-iCKO mouse lenses compared to Ctrl mouse lenses at P17-19 according to RNA-seq (*p* < 0.05; right-tailed Fischer's exact *t*-test). The top enriched downregulated canonical cellular pathways are ranked according to *p*-value. Exact p-value is listed in Supplementary Table 2. Cellular pathways involved in cholesterol biosynthesis are labeled in red. **b** Heatmap showing the relative expression value (Z score) of all 19 genes involved in cholesterol biosynthesis in Ctrl (*n* = 3) and *Qk*-iCKO (*n* = 3) lenses according to RNA-seq data ranked by fold change. **c** Cholesterol biosynthesis pathway with the genes encoding the enzymes involved in each step of the pathway. Red labeled indicate the cholesterol biosynthesis genes downregulated shown in **b**. **d** Results of RT-qPCR analysis of cholesterol biosynthesis genes in isolated Ctrl (*n* = 3) and *Qk*-iCKO (*n* = 3) lenses at P17-19. *p* = 0.0001552 (Hmgcs1); 0.001142 (Hmgcr); 0.002307 (Pmvk); 0.000008831 (Mvd); 0.0005463 (Idi1); 0.00003504 (Fdps); 0.0009219 (Fdft1); 0.00001013 (Lss); 0.00003438 (Cyp51); 0.00002009 (Tm7sf2); 0.000001402 (Msmo1); 0.000006230 (Nsdhl); 0.00003707 (Hsd17b7); 0.00003894 (Sc5d), **p* < 0.01; ***p* < 0.001; ****p* < 0.0001 (two-tailed unpaired *t*-test). The results are presented as means with SD. **e, f** Immunoblots and quantification of enzymes involved in cholesterol biosynthesis (Hmgcs1, Hmgcr, and Fdps) and immunoblots of Qki-5 and Qki-6 in isolated Ctrl (*n* = 3) and *Qk*-iCKO (*n* = 3) lenses at P19. β-actin: loading control. *p* = 0.009136 (Hmgcs1); 0.02481 (Hmgcr); 0.0006804 (Fdps), **p* < 0.05; ***p* < 0.01; ***p* < 0.001 (two-tailed unpaired *t*-test). The results are presented as means with SD. **g** Schematic of the differentiation of NSCs to NLPCs (right). Bright-field images and immunostains for Pax6 (green) and αB-crystallin (red) in NSCs and NLPCs (left). DAPI (blue): nuclei. Scale bar: 50 μm. **h** Immunofluorescent staining of WT and Qki-depleted (*Qk*^-/-^) NSCs for Qki-5. DAPI (blue): nuclei. Scale bar: 50 μm. **i** The top enriched downregulated canonical cellular pathways in *Qk*^-/-^ NLPCs (*n* = 3) compared to WT NLPCs (*n* = 3) according to RNA-seq (IPA; *p* < 0.05; right-tailed Fischer's exact *t*-test) are ranked according to *p*-value. Exact *p*-value is listed in Supplementary Table 3. Cellular pathways involved in cholesterol biosynthesis are labeled in red. **j** Heatmap of the relative average expression value (Z score) of all 19 genes involved in cholesterol biosynthesis according to RNA-seq data in WT (*n* = 3) and *Qk*^-/-^ (*n* = 3) NLPCs and WT (*n* = 3) and *QKI* KO (*n* = 3) HLE-B3 cells. **k** Results of RT-qPCR analysis of cholesterol biosynthesis genes in WT (*n* = 3) and *Qk*^-/-^ (*n* = 3) NLPCs. *p* = 0.01012 (Hmgcs1); 0.02979 (Hmgcr); 0.003582 (Mvd); 0.0009995 (Idi1); 0.00006936 (Fdps); 0.001929 (Cyp51); 0.03582 (Msmo1); 0.001525 (Nsdhl); 0.0000009839 (Sc5d); 0.02260 (Dhcr24), **p* < 0.05; ***p* < 0.01; ***p* < 0.001; ****p* < 0.0001 (two-tailed unpaired *t*-test). The results are presented as means with SD. All the experiments were replicated three times in the lab.

the eyes of *Qk*-iCKO mice for 1 week (from P14 to P21), during which the cataract phenotype developed rapidly (Fig. 3d)[19,56,57]. We found that lanosterol significantly reduced the opacity and improved the transparency of *Qk*-deficient lenses (Fig. 3e, f), suggesting that regulation of sterol levels by Qki is required for maintaining the transparency of the lens. Taken together, these data suggested that Qki is a transcriptional regulator of cholesterol biosynthesis in eye lens cells and that cataracts induced by Qki depletion were caused by insufficient cholesterol biosynthesis.

**QKI-5 cooperates with SREBP2 to regulate transcription of cholesterol biosynthesis genes.** We found that the nuclear localized splicing isoform of QKI, QKI-5 was most abundant in chromatin fractions compared to the nuclear soluble fractions in HLE-B3 cells (Fig. 4a), and our lab recently showed that Qki-5 functions as a transcriptional co-activator of peroxisome proliferator-activated receptor beta (PPARβ)-retinoid X receptor alpha (RXRα) complex[58], suggesting that QKI-5 might also regulate the expression of genes involved in cholesterol biosynthesis. Besides, Srebp2, the major transcription factor of cholesterol biosynthesis genes was one of the top upstream transcription regulators of differentially expressed genes in the lenses from control and *Qk*-iCKO mice, including all of the cholesterol biosynthesis genes regulated by Qki, according to IPA (Supplementary Fig. 4g). In addition, it has been shown that mutations of SREBP2 target genes in human and *Srebf2* (the gene encoding Srebp2) mutations in mouse led to cataract formation[19,25,31,32], recapitulating the phenotype of *Qk*-iCKO mice. Therefore, we next sought to determine whether QKI-5 regulates expression of the genes involved in cholesterol biosynthesis by modulating SREBP2-mediated cholesterol biosynthesis. We first reasoned that chromatin-bound QKI-5 might form a complex with the mature form of SREBP2, which is localized to the nucleus to activate transcription. Under cholesterol withdrawal, when SREBP2 activity is transcriptionally robust to drive de novo cholesterol biosynthesis, we found that QKI-5 interacted with mature SREBP2 both endogenously and ectopically as demonstrated in HLE-B3 cells via co-immunoprecipitation (co-IP) (Fig. 4b, c and Supplementary Fig. 4h, i). We compared the interaction of QKI-5 and mature SREBP2 from the nuclear lysate of HLE-B3 cells

under the sterol depletion condition to that under the sterol repletion condition[59]. Interaction between QKI-5 and mature SREBP2 was reduced in sterol-repleted cells in IPs with both anti–Qki-5 and anti-Srebp2 antibodies (Supplementary Fig. 4h, i), which suggests that interaction of QKI-5 with mature SREBP2 can be induced by a sterol-low condition to meet the high demand of cholesterol biosynthesis in the lens cells. Interestingly, we observed the decreased protein levels of QKI-5 and mature SREBP2 in the nuclear lysate input from the sterol-repleted HLE-B3 cells (Supplementary Fig. 4h, i), suggesting that the decreased nuclear localization of SREBP2 and/or QKI-5 expression/stability might be the other factors that contribute to the reduced binding between QKI-5 and mature SREBP2 in the lens cells. Consistently, Qki-5 and Srebp2 also formed a complex in NLPCs (Fig. 4d, e). Besides QKI-5, we also tested if another QKI isoform, QKI-6, which can be localized to the nucleus in a cell-type specific manner[38], can interact with SREBP2. QKI-6 was abundantly expressed in the nucleus compared to the cytosol in HLE-B3 cells (Supplementary Fig. 4j), consistent with the observation in the lens tissue indicated by the staining with anti–Qki-6 antibody (Supplementary Fig. 1e). Different from QKI-5, which is predominately localized on the chromatin, QKI-6 was similarly distributed between nuclear soluble and chromatin fractions (Supplementary Fig. 4j). In addition, we further showed that interaction of QKI-6 with SREBP2 was very minimal (Supplementary Fig. 4h, i), suggesting that QKI-5 is the major QKI isoform cooperating with SREBP2 in lens cells.

We further asked whether the complex of Qki-5 and Srebp2 regulates transcription in lens cells. To answer this question, we performed chromatin IP followed by high-throughput DNA sequencing (ChIP-seq) with NLPCs to determine whether Qki-5 and Srebp2 co-localize to promoter regions genome-wide. We found that 40.36% of Qki-5 peaks, 64.53% of Srebp2 peaks, and 46.12% of RNA polymerase II (Pol II) peaks in NLPCs were in the promoter regions, whereas promoter regions accounted for only 4.41% of the reference genome (Fig. 4f), demonstrating that Qki-5 preferentially interacts with gene promoters as do Srebp2 and Pol II. We also showed that in HLE-B3 cells, QKI-5 (23.20%) and SREBP2 (22.60%) binding events were highly enriched in the promoter regions, whereas total promoter regions only account for 1.80% of the entire genome (Supplementary Fig. 5a).

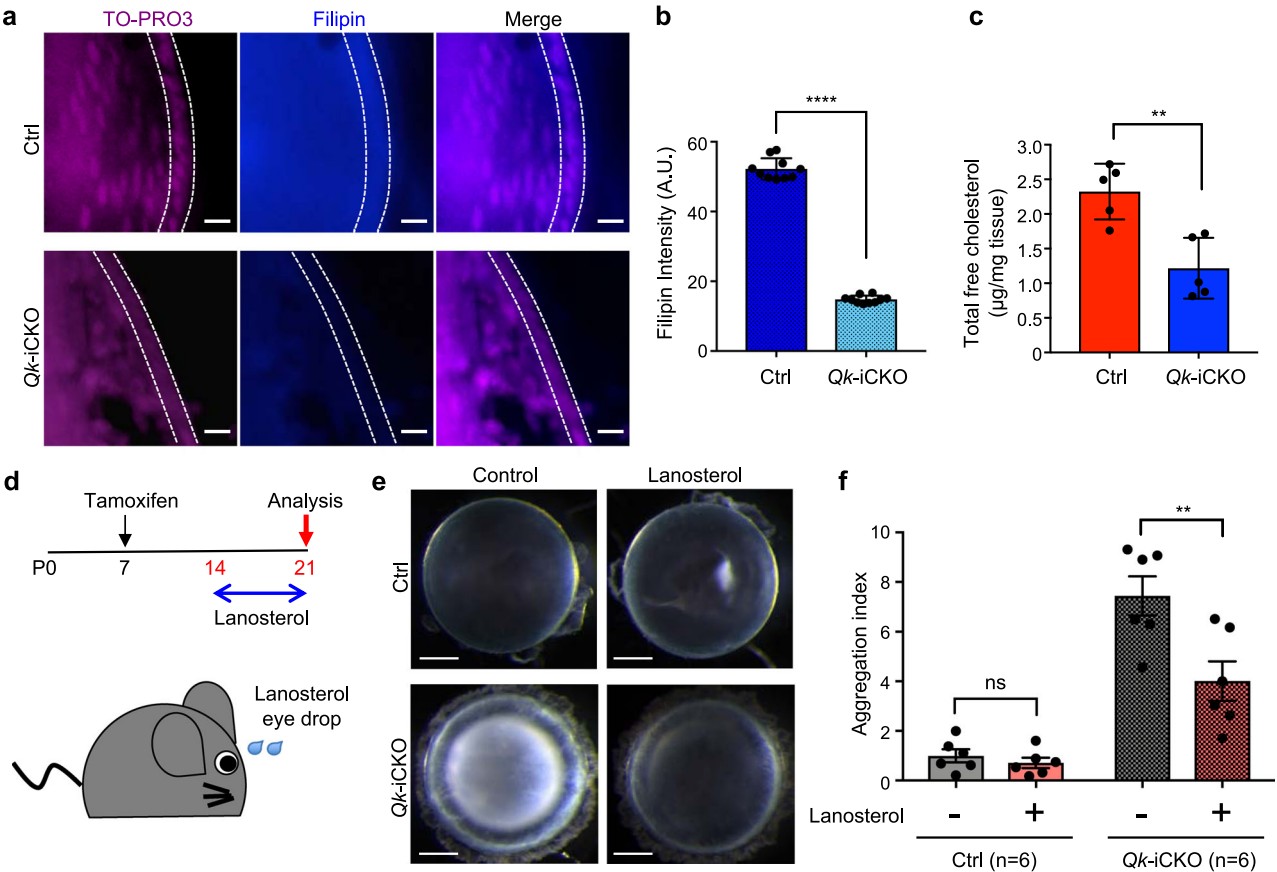

**Fig. 3 Cholesterol levels are decreased in Qki-depleted lenses, and sterol supply alleviates the cataract phenotype in the lenses of *Qk*-iCKO mice.**
**a** Filipin (blue) staining of lens cells from frozen sections of the eyes of Ctrl and *Qk*-iCKO mice at P19. TO-PRO3 (magenta): nuclei. Scale bar: 10 μm.
**b** Quantification of filipin intensity (A.U.) in the regions between the white dotted lines in Ctrl and Qki-depleted lenses shown in **a**. ****$p < 0.0001$ (two-tailed unpaired $t$-test). The results are presented as means with SD. 10 cells from the images shown in **a** are counted for quantification. Each dot represents the intensity of individual cells. **c** Quantification of total free cholesterol levels (μg/mg tissue) in lenses isolated from Ctrl ($n = 5$) and *Qk*-iCKO ($n = 5$) mice at P19. $p = 0.0032$, **$p < 0.01$ (two-tailed unpaired $t$-test). The results are presented as means with SD. **d** Timeline and schematic of lanosterol-based treatment in mice. **e** Representative images of lenses isolated from Ctrl and *Qk*-iCKO mice given either mock (control) or lanosterol-based treatment for 1 week. Scale bar: 0.5 mm. **f** Quantification of the area of protein aggregates from isolated Ctrl ($n = 6$) and *Qk*-iCKO ($n = 6$) lenses after mock and lanosterol-based treatment. Aggregation index = x/average of the area of aggregates in Ctrl lenses with the mock treatment. x = area of aggregates. $p = 0.1668$ (Ctrl); 0.0014 (*Qk*-iCKO), **$p < 0.01$; ns not significant ($p \geq 0.05$) (two-tailed paired $t$-test). The results are presented as means with standard error of the mean (SEM). All the experiments were replicated three times in the lab.

Furthermore, we found that the QKI-5, SREBP2, and POL II peaks were greatly enriched and correlated with at transcription start sites (TSSs) in both NLPCs and HLE-B3 cells (Fig. 4g, h, Supplementary Fig. 5b, c). These data suggest the cooperation between QKI-5 and SREBP2 in regulation of transcription.

Reinforcing the notion that Qki-5 is involved in the regulation of transcription of Srebp2 targets, we found that 9,032 promoters were co-occupied by Qki-5, Srebp2, and Pol II, which accounted for 91.84% of the total promoters occupied by Srebp2 ($n = 9,834$) in NLPCs (Fig. 5a). Similarly, SREBP2 highly co-occupied the promoter regions with QKI-5 in HLE-B3 cells, as 76.77% of the SREBP2-binding promoters (152 of 198) were co-bound by QKI-5 and POL II (Supplementary Fig. 5d). Additionally, the known DNA-binding motif of SREBP2 was enriched in the QKI-5 binding peaks in both HLE-B3 cells and NLPCs (Fig. 5b), indicating a high likelihood of transcriptional regulation of QKI-5 in the SREBP2 target genes. Taken together, these data suggested that QKI-5 regulates SREBP2-mediated transcription in eye lens cells.

To further identify Srebp2-bound genes whose transcription is specifically regulated by Qki-5, we performed cellular pathway enrichment analysis of the overlapping promoter-bound peaks

($n = 9,032$) in Qki-5, Srebp2, and Pol II ChIP-seq data in NLPCs (Fig. 5a). We found that cholesterol biosynthesis was the most enriched pathway determined by IPA (Fig. 5c) and that 12 of the 19 cholesterol biosynthesis enzymes were enriched as Qki-5/Srebp2/Pol II targets (Fig. 5d). Additionally, we showed that Qki-5, Srebp2, and Pol II were colocalized to the promoter regions of all 19 cholesterol biosynthesis genes, visualized as individual tracks in ChIP-seq data (Fig. 5e, Supplementary Fig. 5e). These data suggested that the Qki-5/Srebp2/Pol II complex predominantly occupies cholesterol biosynthesis genes. To further validate direct target genes of Qki-5, we clustered the genes ($n = 301$) whose promoters are occupied by Qki-5 and expressions are significantly downregulated in both *Qk*-iCKO lenses and *Qk*-deficient NLPCs according to RNA-seq data (Fig. 5f). We found that the cholesterol biosynthesis pathway was the most enriched pathway determined by IPA (Fig. 5g), suggesting that transcription of cholesterol biosynthesis genes is directly regulated by Qki-5 in lens cells. Of note is that 10,329 genes bound by Qki-5 were not downregulated, suggesting the existence of additional transcriptional mechanisms of tissue-specific Qki-5–mediated gene expressions (Supplementary Fig. 5f). Because QKI-5 also interacts with promoters of other genes in addition to SREBP2 target genes, we

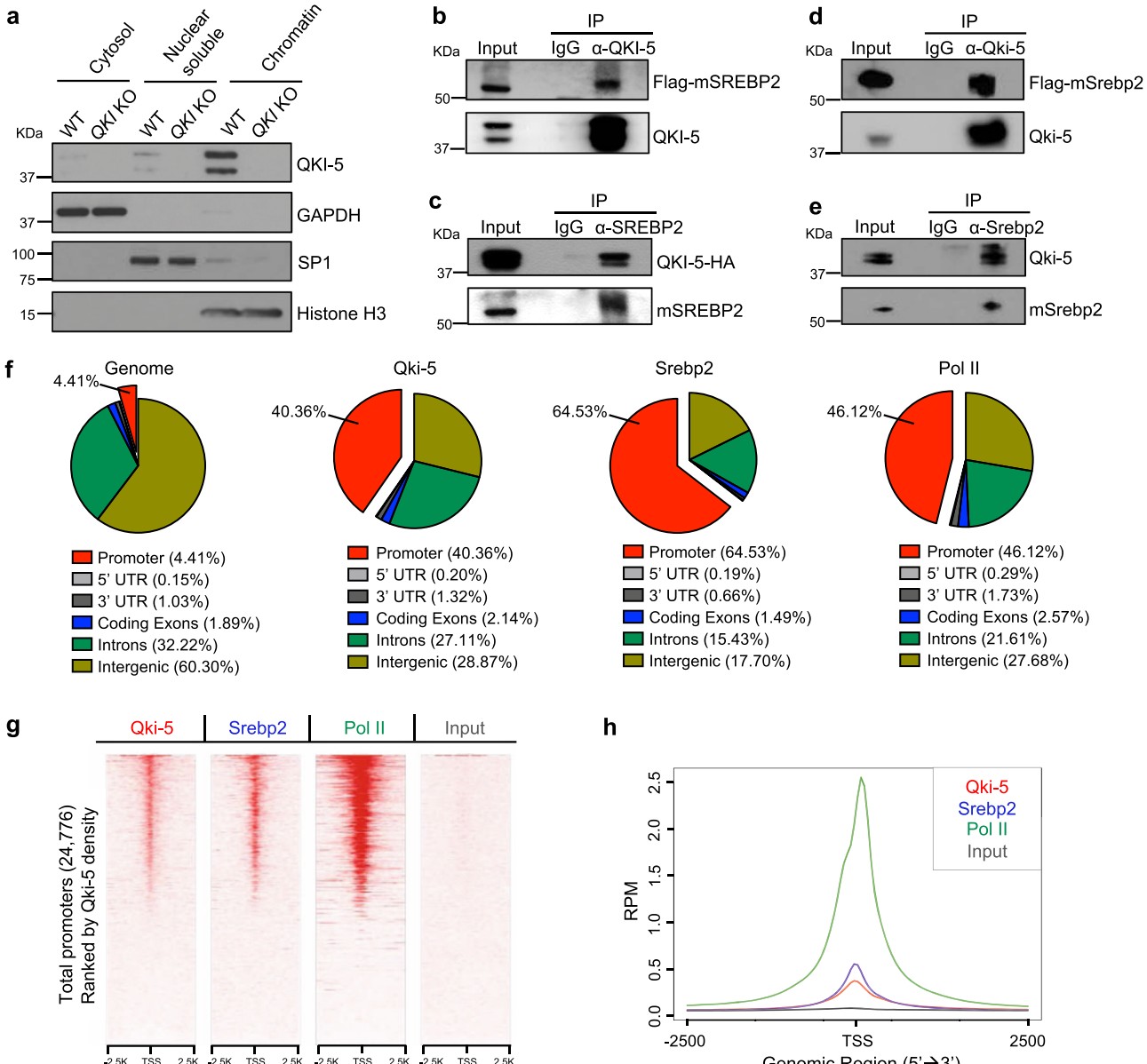

**Fig. 4 QKI-5 cooperates with SREBP2 at promoters genome-wide in lens cells. a** Immunoblots of QKI-5 in subcellular fractions of WT and *QKI* KO HLE-B3 cells. GAPDH: cytosol; SP1: nuclei; Histone H3: chromatin. The result was repeated three times. **b**, **c** Co-IP of HLE-B3 cells overexpressing QKI-5–HA and Flag-SREBP2 with **b** anti–QKI-5 antibody blotting with anti-Flag and –QKI-5 antibodies and **c** anti-SREBP2 antibody blotting with anti-HA and -SREBP2 antibodies. mSREBP2: mature SREBP2. The results were repeated three times. **d**, **e** Co-IP of NLPCs overexpressing Flag-Srebp2 with **d** anti–Qki-5 antibody blotting with anti-Flag and –Qki-5 antibodies and **e** anti-Srebp2 antibody blotting with anti–Qki-5 and -Srebp2 antibodies. The results were repeated three times. **f** Genomic annotation of genome or Qki-5–, Srebp2-, and Pol II-binding sites from ChIP-seq in NLPCs. Promoters are confined to the regions of ±2 kb from the TSS. **g** Heatmap of Qki-5, Srebp2, and Pol II ChIP-seq and input signals within ±2.5 kb from the TSS regions for all promoters ($n = 24,776$) in NLPCs ranked by Qki-5 density. **h** Signalplots showing the read counts per million mapped reads (RPM) for Qki-5, Srebp2, and Pol II ChIP-seq data and input ±2.5 kb from the TSS of all promoter regions ($n = 24,776$) in NLPCs.

asked whether QKI-5 has a stronger feature of co-occupancy with SREBP2 than with other transcription factors in lens cells. To that end, we stratified the Qki-5 ChIP-seq read densities (RPM) based on whether they overlapped with Srebp2 ChIP-seq peak densities (RPM) in NLPCs (Fig. 5h). We found that the Qki-5 ChIP-seq RPM on the promoter regions of Qki-5+Srebp2+–bound genes ($n = 9,047$) were considerably higher than those on the promoter regions of Qki-5+Srebp2−–bound genes ($n = 1,583$) (Fig. 5h), a phenomenon also observed in HLE-B3 cells (Supplementary Fig. 5g). Taken together, these data suggested that QKI-5 preferentially binds to SREBP2-associated promoters over other promoters.

**QKI transcriptionally enhances cholesterol biosynthesis by facilitating SREBP2/POL II recruitment.** Based on the fact that QKI-5 and SREBP2 formed a complex and co-occupied the promoters of the genes involved in the cholesterol biosynthesis pathway, which was reflected by the downregulation of genes during cholesterol biosynthesis upon QKI depletion, we sought to test the hypothesis that QKI regulates transcription of cholesterol biosynthesis by enhancing the recruitment of SREBP2 to the promoter regions of cholesterol biosynthesis genes. Indeed, we found that the association of mature SREBP2 with chromatin was reduced in *QKI*-deficient HLE-B3 cells (Fig. 6a). Also, consistent

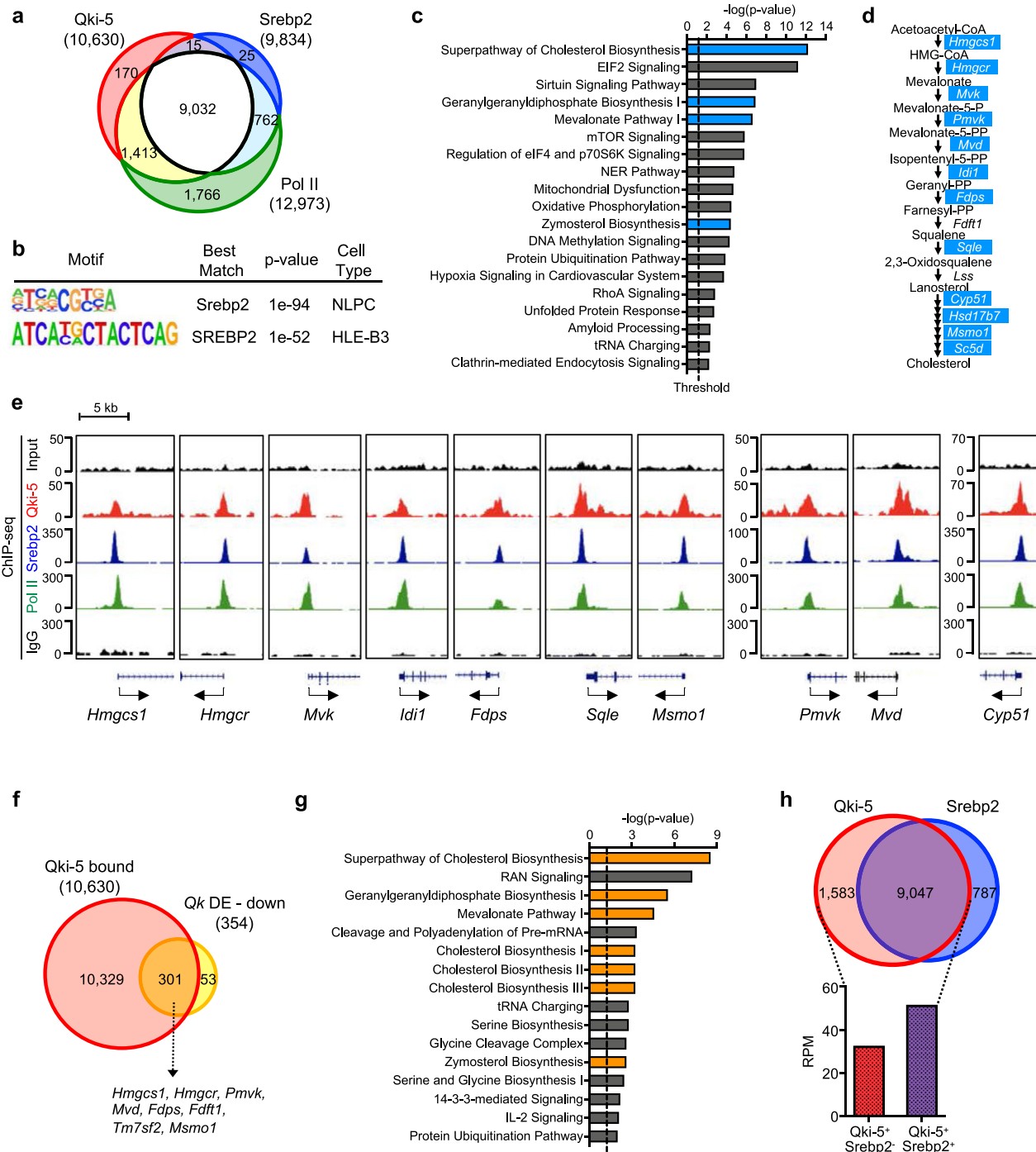

**Fig. 5 Qki-5 cooperates with Srebp2 to regulate transcription of cholesterol biosynthesis genes. a** Venn diagram showing the number of overlapping promoters bound by Qki-5, Srebp2, and Pol II from ChIP-seq in NLPCs. Promoters defined by TSS ± 2 kb. **b** DNA-binding motif similar to the known SREBP2 motif was enriched in QKI-5 ChIP-seq peaks in NLPCs and HLE-B3 cells using HOMER motif analysis. *p*-value was derived by HOMER using cumulative binomial distributions. **c** IPA of the top 1,000 genes (ranked according to RPM within ±0.5 kb of TSS in Srebp2 ChIP-seq) in overlapping binding promoters shown in **a** (*n* = 9,032). Canonical cellular pathways are ranked according to significance (*p*-value) (*p* < 0.05; right-tailed Fischer's exact *t*-test). The blue labels the canonical cellular pathways including cholesterol biosynthesis genes. **d** Cholesterol biosynthesis pathway and genes encoding the enzymes involved in the cholesterol biosynthesis. Blue-labeled genes are clustered in the cholesterol biosynthesis pathway in **c**. **e** UCSC Genome Browser snapshot of the promoter regions of cholesterol biosynthesis genes encoding the cholesterol biosynthesis enzymes labeled in blue in **d**, which are co-bound by Qki-5, Srebp2, and Pol II in NLPCs. Input and rabbit IgG are used as controls. **f** Venn diagram showing the overlapping genes between Qki-5–bound genes from Qki-5 ChIP-seq data in NLPCs and downregulated genes in both *Qk*⁻/⁻ NLPCs and *QKI* KO HLE-B3 cells relative to WT according to RNA-seq (*p* < 0.05; two-tailed Wald test). DE: differentially expressed. **g** IPA of the overlapping genes in **f** (*n* = 301) ranked according to significance (*p*-value) (*p* < 0.05; right-tailed Fischer's exact *t*-test). Cellular pathways involved in cholesterol biosynthesis are labeled in orange. **h** Comparison of the average RPM values of the Qki-5⁺Srebp2⁻ (*n* = 1,583) and Qki-5⁺Srebp2⁺ (*n* = 9,047) ChIP-seq binding events within ±0.5 kb from the TSS shown in NLPCs.

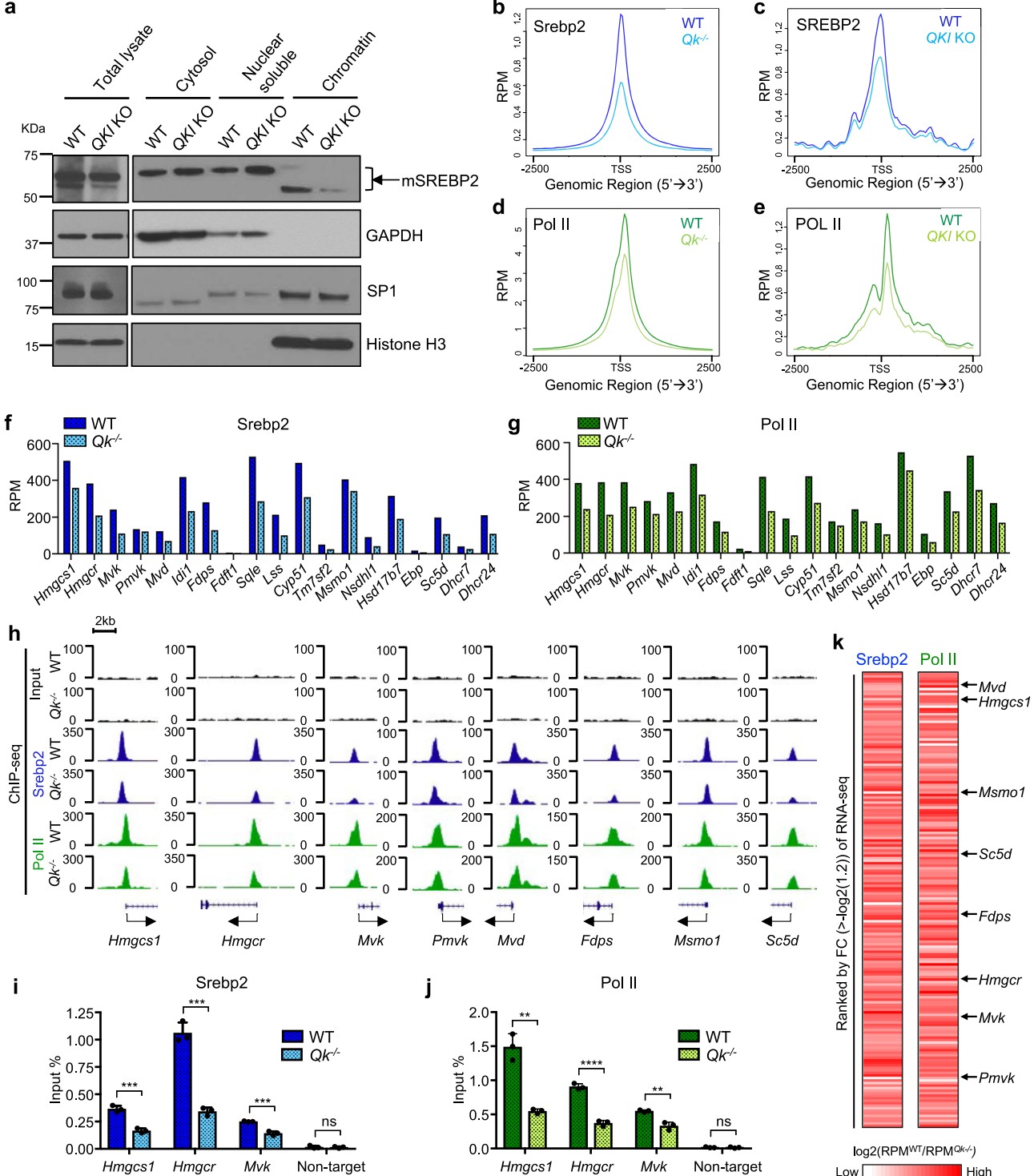

**Fig. 6 QKI-5 transcriptionally enhances cholesterol biosynthesis by facilitating SREBP2/POL II recruitment. a** Immunoblots for detection of mSREBP2 (mature SREBP2) in subcellular fractions of HLE-B3 cells. GAPDH: cytosol; SP1: nuclei; Histone H3: chromatin. The result was repeated three times. **b**, **c** Signalplots showing RPM for SREBP2 in **b** NLPCs and **c** HLE-B3 cells ±2.5 kb from the TSS of SREBP2-bound peaks in WT and *QKI* KO cells. **d**, **e** Signalplots showing the read counts per million mapped reads for POL II in **d** NLPCs and **e** HLE-B3 cells ±2.5 kb from the TSS of SREBP2-bound peaks in WT and *QKI* KO cells. **f**, **g** RPM value of all the 19 cholesterol biosynthesis genes in **f** Srebp2 ChIP-seq and **g** Pol II ChIP-seq in WT and *Qk*−/− NLPCs. **h** Representative UCSC Genome Browser snapshot of the promoter regions of cholesterol biosynthesis genes from ChIP-seq of NLPCs. **i**, **j** Srebp2 and Pol II ChIP-qPCR analysis of the promoter regions of *Hmgcs1*, *Hmgcr*, and *Mvk* in WT and *Qk*−/− NLPCs. **i** $p = 0.0006554$ (*Hmgcs1*); 0.0002616 (*Hmgcr*); 0.0005623 (*Mvk*); 0.7637 (Non-target). **j** $p = 0.001092$ (*Hmgcs1*); 0.00009184 (*Hmgcr*); 0.002848 (*Mvk*); 0.7947 (Non-target). **p < 0.01; ***p < 0.001; ****p < 0.0001; ns not significant ($p \geq 0.05$) (two-tailed unpaired *t*-test). The results are presented as means with SD. Data are representative of three independent experiments. **k** Heatmap of normalized fold change of RPM in WT NLPCs over *Qk*−/− NLPCs (log2(RPM^WT^/RPM^Qk-/-^) in Srebp2 and Pol II ChIP-seq data ranked by fold change in RNA-seq of NLPCs. Arrows indicate cholesterol biosynthesis genes.

with previous studies, we observed two different sizes of mature SREBP2[60–64], and the lower molecular weight of these two sizes of mature SREBP2 was highly associated with chromatin (Fig. 6a). Our finding of significant reduction of chromatin-associated SREBP2 upon QKI depletion in HLE-B3 cells suggests that QKI helps recruit SREBP2 to promoters or stabilize it on promoters. Supporting this notion, SREBP2 ChIP-seq data demonstrated that occupancy by SREBP2 in the promoter regions of SREBP2-bound genes decreased in NLPCs and HLE-B3 cells upon QKI depletion (Fig. 6b, c). Consistently, POL II occupancy in the promoter regions of SREBP2-bound genes also decreased with QKI depletion in both NLPCs and HLE-B3 cells (Fig. 6d, e), implying that recruitment of the transcription complex SREBP2/POL II was reduced by QKI depletion genome-wide. Consistent with our finding that cholesterol biosynthesis was one of the most enriched cellular pathways directly regulated by Qki-5, the bindings of both Srebp2 and Pol II to the cholesterol biosynthesis genes were significantly decreased upon Qki depletion in NLPCs (Fig. 6f–h, Supplementary Fig. 6a). ChIP-qPCR also confirmed the reduced Srebp2 and Pol II recruitment to the promoter regions of *Hmgcs1*, *Hmgcr*, and *Mvk* upon Qki depletion in NLPCs (Fig. 6i, j). We further revealed that the reductions of Srebp2 and Pol II binding on the promoters of the genes highly bound by Srebp2 were significantly correlated with the down-regulations of gene expressions upon Qki depletion as cholesterol biosynthesis genes such as mevalonate diphosphate decarboxylase (*Mvd*), *Hmgcs1*, methylsterol monooxygenase 1 (*Msmo1*), *Sc5d*, *Fdps*, *Hmgcr*, *Mvk*, and phosphomevalonate kinase (*Pmvk*) all fell into this gene cluster as shown in Fig. 6k, suggesting that Qki-5 is essential for transcriptional enhancement of these cholesterol biosynthesis genes. Consistently, cholesterol biosynthesis genes, including *FDPS*, squalene epoxidase (*SQLE*), *LSS*, *MVD*, *MVK*, cytochrome P450 family 51 (*CYP51*), *HMGCS1*, *HMGCR*, *MSMO1*, and hydroxysteroid 17-beta dehydrogenase 7 (*HSD17B7*), were among the 152 genes (Supplementary Fig. 6b) whose SREBP2 and POL II peaks were reduced most dramatically upon *QKI* loss in HLE-B3 cells (Supplementary Fig. 6b–e). Consistently, ChIP-qPCR confirmed the reductions of SREBP2 and POL II on the promoter regions of *HMGCR*, *MVD*, *FDPS*, and *CYP51* upon QKI depletion in HLE-B3 cells (Supplementary Fig. 6f, g). Of note, chromatin occupancy of Pol II was not exclusively dependent on the presence of Qki-5 for non-cholesterol biosynthesis genes as shown in Supplementary Fig. 7a. Reinforcing the critical role of QKI-5 for recruiting and activating SREBP2 on the promoters of cholesterol biosynthesis genes, the transcriptional activity of SREBP2 was significantly enhanced by ectopic expression of QKI-5 (Supplementary Fig. 7b), which suggests that QKI-5 serves as a co-activator of SREBP2. Taken together, these data suggested that QKI-5 facilitates formation of SREBP2-mediated transcriptional complexes in the cholesterol biosynthesis gene promoters and enhances the transcription of these genes.

**Qki interacts with single-stranded DNA.** The major functional domain of Qki is the STAR (signal transduction activator of RNA metabolism) domain, which includes QUA1, K homology (KH), and QUA2 components, and it has been shown that the KH domain binds to both RNA and single-stranded DNA (ssDNA)[65,66]. For instance, both heterogeneous nuclear ribonucleoprotein K (hnRNP K) and far upstream element-binding protein 1 (FUBP1) are known to bind to ssDNA and RNA through the KH domain[67,68]. Accordingly, we hypothesized that QKI might also bind to ssDNA to regulate SREBP2-dependent transcription. To determine whether QKI binds to DNA, we purified recombinant full-length Qki-5 and the STAR domain of Qki-5 protein from *Escherichia coli* and tested

whether they interact with DNA and RNA in vitro (Supplementary Fig. 8a). Using microscale thermophoresis (MST) and isothermal titration calorimetry (ITC) assays, we confirmed that purified full-length and the STAR domain of Qki-5 bind to two known RNA sequences, QRE1 (CUUCUUAAUAUAACUGCCUUAAA-CUUUAAU) and QRE2 (UUCACUAACAA) that contain the QKI-RNA recognition element (QRE) YUAAY (Fig. 7a, Supplementary Fig. 8b, c)[14,66]. Previous studies showed that mutations of UAA can dramatically disrupt QKI-RNA interaction[14,69]. We reasoned that if QKI interacts with ssDNA, it may recognize a similar sequence containing TAA. Because SREBP2-binding motif was enriched in QKI-5 binding peaks both in HLE-B3 and NLPCs (Fig. 5b), we then designed ssDNA oligos originated in the promoter regions of SREBP2 target genes and proximally downstream of the sterol regulatory elements (SREs)[70,71]. These ssDNA oligos were denoted as QKI-DNA recognition elements (QDEs), which all contain TAA (Fig. 7b, c)[14,66]. We found that Qki-5 interacted with QDE sequences derived from the cholesterol biosynthesis genes, such as squalene epoxidase (QDE1), *MVK* (QDE2), *HMGCR* (QDE3), *Hmgcs1* (QDE4), and *FDPS* (QDE5), in the MST assay (Fig. 7c, d). Mutation of a core binding motif (TAA -> GCG) in QDE1-5 led to significant reduction of the binding affinity with Qki-5, suggesting a motif-favorable feature in binding of Qki-5 to DNA. Previous studies showed that the STAR domain of QKI interacts with RNA[66], and consistently, the ITC assay demonstrated that both full-length and the STAR domain of Qki-5 interacted with QDE1 (Supplementary Fig. 8b, c). Since hnRNP K is known to interact with both RNAs and DNAs[72], we then sought to compare the affinities of QKI and hnRNP K in terms of binding to DNAs. To that end, we first purified the KH3 domain of hnRNP K (whose structure has been previously solved by X-ray crystallography)[72] and then tested for the interaction with its known binding motif CTCCCC with both MST and ITC assays. We found that KH3 domain of hnRNP K showed a $K_d = 11.9\,\mu M$ in the MST assay and a $K_d = 13.8\,\mu M$ in the ITC assay (Supplementary Fig. 9a, b), which are both comparable to the binding affinities between the KH domain of Qki and QDE1 in the ITC assay ($K_d = 11.9\,\mu M$) (Supplementary Fig. 8c). Consistently, previous studies showed that hnRNP K binds to single-stranded nucleic acids in the range of $1–3\,\mu M$[72], which is similar to the affinity of Qki measured with the MST assay in this study. Taken together, these data suggested that other than interacting with RNAs, Qki may also enhance Srebp2-mediated transcription potentially by interacting with ssDNAs at promoters.

As QKI-5 forms a molecular complex with SREBP2 in the nucleus (Fig. 4b–e, Supplementary Fig. 4h, i), we further asked if DNA-binding activity of Qki is also dependent on SREBP2 similar to the manner of SREBP2's dependency on QKI-5. Using the *SREBF2*-deficient line of HLE-B3 cells, which displayed the most efficient reduction of SREBP2 expression (Supplementary Fig. 10a), we showed that the genome-wide chromatin localization of QKI-5 was not changed upon SREBP2 depletion (Supplementary Fig. 10b, c), suggesting that SREBP2 is dispensable for QKI-5 binding to chromatin at the whole genomic level. Of note, those genes that are co-targeted by QKI-5 and SREBP2 such as *MVD*, *CYP51*, and *FDPS*, are selectively reduced compared to the QKI-5 target genes that are not bound by SREBP2 (Supplementary Fig. 10d). Taken together, these data suggested that the recruitments of QKI-5 and SREBP2 to the promoters of cholesterol biosynthesis genes might be co-dependent.

## Discussion

Whereas the importance of de novo cholesterol production in eye lens cells is evident in cataracts associated with human hereditary diseases and genetic mouse models with mutations of cholesterol biosynthesis genes[19,25,32], how cholesterol biosynthesis is

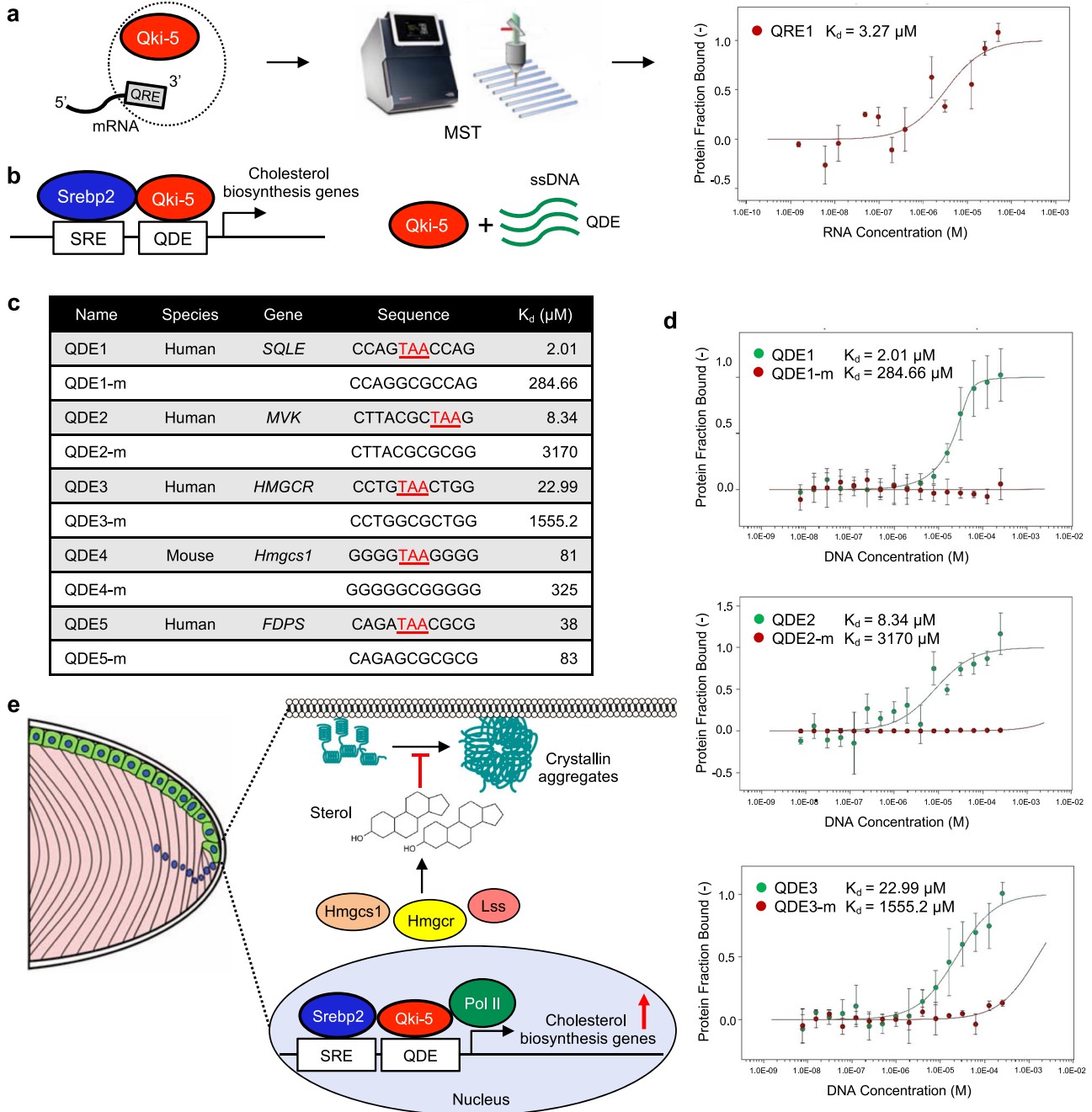

**Fig. 7 Qki-5 interacts with ssDNA. a** Schematic of the MST assay of bacterially expressed recombinant Qki-5 and QRE. The image of MST machine (Monolith NT.115) is from https://nanotempertech.com/monolith. The titration curve for the MST assay for fluorescently labeled Trx-His$_6$–Qki-5 (full-length) and QRE1 (CUUCUUAAUAUAACUGCCUUAAACUUUAAU) is shown at right ($n = 3$). The results are presented as means with SD. **b** Schematic for designing the QDEs proximally downstream of SREs in the promoter regions of cholesterol biosynthesis genes used for the MST assay to examine the interaction with Qki-5. **c** The ssDNA oligos (QDE and QDE-m [mutant]) tested in the MST assay and the K$_d$ value of each oligo in the assay. **d** Titration curves for the MST assay showing the K$_d$ values for fluorescently labeled Trx-His$_6$-Qki-5 (full-length) (fraction of 20 nM protein bound by DNA, y-axis) and QDE/QDE-m (2-fold serially diluted 16 different concentrations (M), x-axis). $n = 3$/group. The results are presented as means with SD. **e** Schematic diagram depicting proposed mechanisms by which Qki-5 cooperates with Srebp2 to enhance transcription of cholesterol biosynthesis, which maintains proper sterol levels to ensure protein homeostasis in eye lens cells and prevent cataracts.

regulated mechanistically in lens cells remained unclear. A low level of intracellular cholesterol triggers translocation of endoplasmic reticulum-embedded SREBP2 to the Golgi apparatus, where SREBP2 is cleaved to a mature form, which is then transported to the nucleus for transcriptional activation of cholesterol biosynthesis genes[73]. Multiple studies have shown that mature SREBP2 interacts with ubiquitous transcription factors such as Sp1 and NF-Y as well as co-activators such as CREB-binding protein to synergistically activate transcription of cholesterol biosynthesis genes[74]. However, studies of the regulation of SREBP2 in a tissue-specific manner are lacking. Our study demonstrated that genetic deletion of *Qk* leads to a cataractagenic

phenotype in the mouse eye lens due to downregulation of the cholesterol biosynthesis pathway. Moreover, we elucidated the mechanisms by which Srebp2-mediated cholesterol biosynthesis is transcriptionally enhanced by Qki to meet the high demand for cholesterol for maintenance of lens transparency. Therefore, we propose that Qki functions as a transcriptional regulator of Srebp2-mediated cholesterol biosynthesis to prevent abnormal protein aggregation in the lens.

Highly enriched cholesterol content in membrane is a unique characteristic of the lens tissue compared to other tissues[15]. Previous studies suggested that sterols are required for the proper function of membrane-embedded α-crystallins, which serve as chaperones to facilitate the folding of lens proteins, including β- and γ-crystallins[18,75,76]. In addition, cholesterol is essential for maintaining membrane integrity, and loss of cholesterol in the cytoplasmic membrane will likely cause influx of $Ca^{2+}$, which has been shown to activate calpain, a protease that aberrantly cleaves crystallins, thereby leading to crystallin aggregation[77]. In our study, we elucidated that Qki plays an essential role in providing sufficient cholesterol in the lens cells. At the tissue level, regulation of cholesterol level by Qki is required for maintaining proper membrane integrity and transparency of the lens tissue. Notably, lens fiber cells undergo differentiation from the outer layer towards center of the lens, and one major process during differentiation is denucleation, which is important for maintaining transparency of the lens[78]. Those terminally differentiated lens fiber cells without organelles are metabolically inactive, which require nutrients/metabolites from the metabolically active neighboring fiber cells. The enclosed fiber cells are tightly packed and behave as a whole similar to a single cell in term of metabolic exchange[79]. Supporting this notion, we showed that AQP0+GFP− cells (lens fiber cells terminally differentiated before the Qki depletion) are also impaired in cellular membrane structure probably due to the indirect impact by Qki-depleted cells (GFP+ cells) (Fig. 1c and Supplementary Fig. 2a), although the impact on the AQP0+GFP− cells was not as remarkable as that on the AQP0+GFP+ cells. Therefore, it is quite likely that Qki-depleted cells potentially affect the neighboring cells by reducing metabolic supports such as cholesterol, which ultimately leads to the massive and fast progression of cataract phenotype. Another potential role of cholesterol in protein homeostasis is to function as an important component of the cell membrane for proper protein trafficking and degradation[80]. Previously, we showed that Qki is an important regulator of endolysosome formation in NSCs[41], so it is also likely that regulation of cholesterol levels by Qki in the eye lens may also contribute to proper trafficking and function of lens proteins for maintaining protein solubility in the lens.

As a KH-domain-containing protein, Qki has been shown to bind to RNA. However, other KH-domain-containing proteins, such as hnRNP K and FUBP1, interact with both RNA and DNA[67]. Here, we demonstrated that, like other KH-domain-containing proteins, Qki-5 can bind to ssDNA (QDE) similar to the RNA-binding motif (QRE) of Qki, which is in close proximity to SREs on the promoters of cholesterol biosynthesis genes[14,66]. Notably, motif analysis on the top peaks of Qki-5 ChIP-seq data by HOMER identified a few known transcription factor-binding motifs (e.g., SREBP2, CAAT box-binding transcription factor/ nuclear factor-1 (CTF/NF1), activating transcription factor-4 (ATF4), nuclear factor erythroid 2 related factor-1 (NRF1), SRY-related HMG-box 10 (SOX10), and PPAR). In addition, QKI-5 binding to the promoters of the cholesterol biosynthesis genes was specifically reduced in the absence of SREBP2 shown in Supplementary Fig. 10d, suggesting the potential contribution of QKI-5–interacting proteins (e.g., SREBP2) to DNA-binding activity of QKI-5 in a context-dependent manner. Therefore, we postulate that the binding of Qki-5 to chromatin is co-determined by its interactions with both DNA and other

transcriptional factors/co-factors[81], although it is also possible that the binding of Qki to chromatin is mediated through RNA. In addition, Qki-5 seemingly shares the feature of various DNA-binding proteins that exhibit weakly conserved sequence specificity, such as the chromatin modifiers such as Chd1 and CHD8[82,83]. Notably, more than one quarter of the Qki-5–bound peaks (28.87% in NLPCs and 32.30% in HLE-B3 cells) fell into intergenic regions in Qki-5 ChIP-seq data (Fig. 4f, Supplementary Fig. 5a). Therefore, Qki-5 potentially activates transcription by stabilizing the transcriptional complex and regulating dynamics of the chromatin structure by interacting with both promoter and cis-regulatory element such as enhancers[84,85]. It is noteworthy to further investigate the spatial distribution of Qki-5 and other key factors in transcriptional regulatory networks and how they coordinate to initiate transcription.

Recent studies suggested that HNRNP K, a well-established KH-domain-containing RNA/DNA-binding protein, played an important role in maintaining epidermal progenitor function through both transcriptional and post-transcriptional regulations[86,87]. Similarly, although we showed that Qki-5 regulates the expressions of cholesterol biosynthesis genes mainly through transcriptional activation, Qki-5 may also regulate some of these genes post-transcriptionally as Qki is known to regulate mRNA metabolism including mRNA splicing, stability, export, and translation[36,88]. In addition to Qki-5, the other two splicing isoforms of Qki, Qki-6, and Qki-7 can potentially contribute to the downregulation of cholesterol biosynthesis genes as our Qk deletion targeted depletion of all three isoforms.

We showed that Qki-5 and Srebp2 bound to the noncanonical genes in addition to the cholesterol biosynthesis genes based on the ChIP-seq data in lens cells, which suggests that Qki-5 may cooperate with Srebp2 in regulating alternative cellular pathways that control protein homeostasis in eye lens cells. Notably, the protein ubiquitination pathway, which is known to regulate protein homeostasis[89], was one of the pathways most enriched in both Qki-5/Srebp2-binding events and downregulated genes in RNA-seq in lens cells (Fig. 2i and Fig. 5c, g). Most abundant ubiquitin-proteasome-related genes decreased upon Qki loss belong to the 26 S proteasome subunits that are involved in degradation of poly-ubiquitinated proteins[90], suggesting that the proteasomal degradation process is compromised with Qki depletion, leading to the accumulation of undegraded poly-ubiquitinated proteins (Supplementary Fig. 3b). As impairment of proteasomal degradation is associated with protein aggregation diseases including neurodegenerative diseases (NDs) and cataracts[91,92], the potential role of Qki in proteasome regulation is of interest for future investigation.

As a universal regulator of cholesterol biosynthesis in many tissues, SREBP2 may not be a feasible therapeutic target to modulate tissue-specific cholesterol levels. Therefore, the molecular mechanism of cholesterol biosynthesis regulation by Qki that we describe herein provides specificity in targeting cholesterol biosynthesis in a tissue-specific manner and opens up possibilities for noninvasive treatment of cataracts. Furthermore, because α-crystallins belong to a family of heat shock proteins known to function as chaperones for coping with aggregation-prone proteins involved in NDs such as tau and α-synuclein[93–95], understanding the role of cholesterol in protein homeostasis in the eye lens will potentially facilitate understanding of protein homeostasis in the brain.

## Methods

**Mice.** *Nestin-CreER*T2 mice (C57BL/6) were gifts from R. Kageyama (Kyoto University, Kyoto, Japan)[96]. *Rosa26-CreER*T2 mice (C57BL/6)[97] and *mTmG* mice (C57BL/6)[47] were from Jackson Laboratories (Bar Harbor, ME). Conditional *Qk* knockout mice (*Qk*L/L) with two *loxP* sequences flanking the exon 2 of *Qk* gene

were generated by our lab as previously described[41]. $Qk^{L/L}$ mice were crossed with *Nestin-CreER*[T2] transgenic mice or *Rosa26-CreER*[T2] mice wherein the expression of tamoxifen-inducible Cre was under the control of the *Nestin* promoter or *Gt (ROSA)26Sor* promoter. Mice were maintained at MD Anderson's animal facility under pathogen-free conditions, maintained under a 12-h light-dark schedule, allowed free access to water and food, and monitored for signs of illness every other day. All mouse experiments were conducted according to the NIH guidelines, and protocols for mouse procedures were approved by the Institutional Animal Care and Use Committee of The University of Texas MD Anderson Cancer Center. *Nestin-CreER*[T2];$Qk^{L/L}$ mice were injected subcutaneously with 20 μL of tamoxifen (10 mg/mL) (Sigma–Aldrich) dissolved in corn oil for two consecutive days at the age of P7 and P8 to induce the deletion of *Qk*. Littermates (*Nestin-CreER*[T2];$Qk^{+/+}$, *Nestin-CreER*[T2]; $Qk^{L/+}$, or $Qk^{L/L}$ mice) of the same age and genetic background were injected concomitantly with tamoxifen and used as controls. The tamoxifen-injected mice were monitored daily to check their health conditions. There were no randomization and blinding events in the animal studies.

**Primary cells**. For NSC isolation, the whole brains of the *Nestin-CreER*[T2];$Qk^{L/L}$ mice at P1 were dissected, sliced into small pieces, and dissociated enzymatically using Neural Tissue Dissociation Kits, according to the manufacturer's instructions (Miltenyi Biotec). The single-cell suspension of NSCs was then maintained in NeuroCult Basal Medium (Stemcell Technologies) containing NeuroCult Proliferation Supplement (Stemcell Technologies), 20 ng/mL EGF (ProteinTech), 10 ng/mL bFGF (ProteinTech), 50 units/mL penicillin G (Thermo Fisher Scientific), and 50 μg/mL streptomycin (Thermo Fisher Scientific). For gene deletion of *Qk*, the NSCs were treated with 100 nM 4-hydroxytamoxifen (Sigma–Aldrich) two times.

For NLPC differentiation, NSCs were seeded onto the culture dishes precoated with 2 ug/ml fibronectin (R&D Systmes) in PBS overnight in the NSC medium described above. After two days, NLPC differentiation was induced in DMEM/F12 (Thermo Fisher Scientific, 11320033) supplemented with 0.05% bovine serum albumin (Sigma–Aldrich), 1% nonessential amino acids (Thermo Fisher Scientific), N-2 (Thermo Fisher Scientific), B-27 (Thermo Fisher Scientific), 50 units/mL penicillin G, and 50 μg/mL streptomycin with treatment of 100 ng/mL Noggin (PeproTech) for 6 days (Step 1) and 100 ng/ml bFGF (ProteinTech), 20 ng/ml BMP4 (PeproTech), 20 ng/ml BMP7 (PeproTech) for 12 days (Step 2)[55]. All cell culture was performed under 5% $CO_2$ atmospheric oxygen, at 37 °C in a humidified incubator.

**Cell lines**. HLE-B3 cells were obtained from ATCC (Cat# CRL-11421) and cultured in Eagle's minimum essential medium (ATCC) containing 20% fetal bovine serum (HyClone), 50 units/mL penicillin G, and 50 μg/mL streptomycin. Lipid depletion in HLE-B3 cells was induced with the treatment of 5% lipoprotein depleted fetal bovine serum (Kalen Biomedical LLC), 10 μM compactin (Sigma–Aldrich), and 50 μM mevalonate (Sigma–Aldrich)[12]. 100 μM MG-132 (Sigma–Aldrich) was treated in HLE-B3 to increase nuclear stabilization of mature SREBP2[98]. All cell culture was performed under 5% $CO_2$ atmospheric oxygen, at 37 °C in a humidified incubator.

**Tissue preparation and immunofluorescence**. Mice were euthanized by $CO_2$ inhalation followed by cervical dislocation. Ocular tissues were immediately dissected, fixed in formalin and embedded in paraffin. For frozen sections, dissected ocular tissues were fixed in 4% paraformaldehyde (PFA), dehydrated in 20% sucrose in PBS, and embedded in optimal cutting temperature compound (OCT). The prepared tissue sections were boiled in citrate buffer (Poly Scientific R&D Corp.) for heat-induced antigen retrieval followed by blocking with 10% horse serum for 1 hr at room temperature. The following primary antibodies were used for staining overnight at 4 °C: anti-Nestin (BD Biosciences, 556309, 1:100), anti-GFAP (BD Biosciences, 556330, 1:200), anti–Qki-5 (immunizing rabbit with a short synthetic peptide [CGAVATKVRRHDMRVHPYQRIVTADRAATGN], Genscript, 1:500), anti–Qki-6 (Sigma–Aldrich, AB9906, 1:500), anti-GFP (Abcam, ab13970, 1:200), anti-AQP0 (Alpha Diagnostic International, AQP01-A, 1:100), anti-Hmgcs1 (Abcam, ab155787, 1:200). The sections were incubated with appropriate Alexa Fluor dye-conjugated secondary antibodies (Thermo Fisher Scientific) for 1 hr at room temperature. For immunofluorescence of NLPCs, cells on fibronectin-coated coverslips were fixed with 4% PFA and treated with 0.1% Triton X-100. After blocking with 1% BSA for 1 hr, the cells were incubated with the following primary antibodies overnight at 4 °C: anti-Pax6 (Abcam, ab5790, 1:200), anti-αB-crystallin (Abcam, ab13496, 1:500), anti-Qki-5 (immunizing rabbit with a short synthetic peptide [CGAVATKVRRHDMRVHPYQRIVTADRAATGN], Genscript, 1:1000).

The samples were then incubated with the corresponding Alexa Fluor conjugated secondary antibodies (Thermo Fisher Scientific) for 1 hr at room temperature and mounted using a VECTASHIELD with DAPI (Vector Laboratories). Fillipin staining was performed on the frozen tissues using cell-based cholesterol assay kit (Abcam, ab133116) based on the manufacturer's instructions, and TO-PRO3 (Thermo Fisher Scientific) was used for nuclear counterstaining. Most immunofluorescence images were captured with Leica DMi8 microscope.

The confocal images in Supplementary Fig. 1d, e and in Supplementary Fig. 2a were captured with Nikon A1R confocal microscope. Fluorescent intensity was measured using Fiji-ImageJ (NIH) and quantified in Fig. 1a, b and Supplementary Fig. 1e, f.

**Silver staining**. The lens tissues of control and *Nestin-CreER*[T2];$Qk^{L/L}$ mice were homogenized in radioimmunoprecipitation (RIPA) lysis buffer (50 mM Tris-HCl [pH 8.0], 150 mM sodium chloride, 1% NP-40, 0.5% sodium deoxycholate, and 0.1% sodium dodecyl sulfate, supplemented with freshly added protease inhibitor cocktail). After centrifugation at $13,000 \times g$ at 4 °C for 10 min, the supernatant was collected and quantified using a DC protein assay (Bio-Rad). 3–5 μg of protein lysates was separated by 4–12% gradient NuPAGE gels (Invitrogen) and visualized by Pierce™ Silver Staining Kit (Thermo Fisher Scientific) according to the manufacturer's instructions.

**Mass spectrometry**. Aggregates from the lenses of *Nestin-CreER*[T2];$Qk^{L/L}$ mice were dissected and solubilized with 8 M urea followed by visualization of proteins using Coomassie blue staining. The strong gel bands were excised and analyzed by Taplin Biological Mass Spectrometry Facility (Harvard Medical School). The excised gel bands were cut into ~1 mm[3] pieces. The samples were reduced with 1 mM DTT for 30 min at 60 °C and then alkylated with 5 mM iodoacetamide for 15 min in the dark at room temperature. Gel pieces were then subjected to a modified in-gel trypsin digestion procedure[99]. Gel pieces were washed and dehydrated with acetonitrile for 10 min. followed by removal of acetonitrile. Pieces were then completely dried in a speed-vac. Rehydration of the gel pieces was with 50 mM ammonium bicarbonate solution containing 12.5 ng/μl modified sequencing-grade trypsin (Promega, Madison, WI) at 4 °C. Samples were then placed in a 37 °C room overnight. Peptides were later extracted by removing the ammonium bicarbonate solution, followed by one wash with a solution containing 50% acetonitrile and 1% formic acid. The extracts were then dried in a speed-vac (~1 hr). For the analysis, the samples were reconstituted in 5–10 μl of HPLC solvent A (2.5% acetonitrile, 0.1% formic acid). A nano-scale reverse-phase HPLC capillary column was created by packing 2.6 μm C18 spherical silica beads into a fused silica capillary (100 μm inner diameter x ~30 cm length) with a flame-drawn tip[100]. After equilibrating the column each sample was loaded via a Famos auto sampler (LC Packings, San Francisco CA) onto the column. A gradient was formed, and peptides were eluted with increasing concentrations of solvent B (97.5% acetonitrile, 0.1% formic acid). As each peptide was eluted they were subjected to electrospray ionization and then they entered into an LTQ Orbitrap Velos Pro ion-trap mass spectrometer (Thermo Fisher Scientific, San Jose, CA). Eluting peptides were detected, isolated, and fragmented to produce a tandem mass spectrum of specific fragment ions for each peptide. Peptide sequences (and hence protein identity) were determined by matching protein or translated nucleotide databases with the acquired fragmentation pattern by the software program, Sequest (Thermo-Finnigan, San Jose, CA)[101].

**Congo Red staining**. Formalin-fixed paraffin sections of ocular tissues were rehydrated, washed, and incubated in Congo Red solution (0.5% Congo Red (Fisher Scientific) in 50% alcohol) for 20 min. The tissue sections were then rinsed and transferred to alkaline alcohol solution (1X sodium hydroxide, 50% alcohol) briefly and rinsed. The sections were then counterstained with hematoxylin for 30 sec, rinsed, and dehydrated[102].

**Electron microscopy**. Lenses from control and *Nestin-CreER*[T2];$Qk^{L/L}$ at the age of P30 were immediately dissected after euthanasia described above and fixed in a solution containing 3% glutaraldehyde and 2% paraformaldehyde in 0.1 M cacodylate buffer (pH 7.3) at 4 °C and processed at the High Resolution Electron Microscopy Facility at MD Anderson. The eye lens tissues were then washed in 0.1 M sodium cacodylate buffer and treated with 0.1% Millipore-filtered cacodylate buffered tannic acid, post-fixed with 1% buffered osmium and stained en bloc with 0.1% Millipore-filtered uranyl acetate. The samples were dehydrated in increasing concentrations of ethanol and then infiltrated and embedded in LX-112 medium. The samples were then polymerized in a 60 °C oven for approximately three days. Ultrathin sections were cut using a Leica Ultracut microtome (Leica, Deerfield, IL) and then stained with uranyl acetetate and lead citrate in a Leica EM Stainer. The stained samples were examined in a JEM 1010 transmission electron microscope (JEOL USA, Inc., Peabody, MA) using an accelerating voltage of 80 kV. Digital images were obtained using an AMT imaging system (Advanced Microscopy Techniques Corp., Danvers, MA).

**Lentivirus production and infection of cells**. The coding DNA sequence region of *Srebf2* was amplified by PCR from pLKO-puro Flag-Srebp2 (addgene #32018)[103] and engineered into a pcDNA vector containing 2X Flag to generate an insert of Srebp2 with 2X Flag at the N-terminus (2X Flag-Srebp2). pLKO-puro Flag-Srebp2 containing 1X Flag at the N-terminus was cut with SalI and NotI to remove the Srebp2, and 2X Flag-Srebp2 from pcDNA was fused with the cut vector to generate Srebp2 expressing vector with 3X Flag at the N-terminus using In-Fusion cloning kit (Takara Bio). The coding DNA sequence region of wild-type with a hemagglutinin (HA) tag at the C terminal was first cloned into pENTR TOPO vector

(Thermo Fisher Scientific) and recombined to the lentiviral vector pInducer20[104] using Gateway LR clonase II enzyme mix (Thermo Fisher Scientific) according to the manufacturer's instructions to generate pInducer20–Qki-5–HA vector. The lentiviruses packaged in HEK293T cells were infected to HLE-B3 cells and NSCs. The cells were then treated with puromycin (InvivoGene) for selection of Flag-Srebp2+ cells and G418 (Life Technologies) for Qki-5–HA+ infected cells. The viable cells were used for further experiments. Doxycycline treatment was performed to induce the expression of Qki-5–HA.

Targeted gene deletion of QKI and SREBF2 in HLE-B3 cells was conducted by CRISPR/Cas9 system using plentiCRISPRv2 plasmid[105,106]. After lentiviral infection, QKI-target HLE-B3 cells were enriched by puromycin (InvivoGene) selection.

**Luciferase reporter assay.** HEK293FT cells were seeded onto 96-well plates followed by co-transfection of pcDNA3.1–2XFLAG–SREBP2 vector (addgene #26807) (15340088) (or the empty vector) and pInducer20–Qki-5–HA vector in a dose-dependent manner (1, 5, 25 ng) with pSynSRE-T-Luc luciferase reporter vector (addgene #60444) (9430668) and Renilla co-reporter vector using lipofectamine 2000 (Thermo Fisher Scientific). After 24 h, co-transfected cells were lysed and measured with firefly luciferase activity and Renilla luciferase activity sequentially using Dual-Glo Luciferase Assay System (Promega) according to the manufacturer's instructions. Renilla luciferase activity was used to normalize the transfection efficiency.

**Immunoblotting and IP.** Lens tissue lysates were prepared as described above. Protein lysates of HLE-B3 cells and NLPCs were also prepared from the RIPA-soluble fraction and measured using a DC protein assay (Bio-Rad). 40–100 μg of proteins was separated on 4–12% gradient NuPAGE gels (Invitrogen) and transferred to a nitrocellulose membrane and probed with anti-ubiquitin (MBL, MK-11-3, 1:500), anti-p62 (CST, 5114, 1:500), anti-Hsp90 (Abcam, ab59459, 1:500), anti-Hmgcs1 (Abcam, ab155787, 1:500), anti-Hmgcr (Abcam, ab174830, 1:500), anti-Fdps (Abcam, ab189874, 1:500), and anti-Qki-5 (immunizing rabbit with a short synthetic peptide [CGAVATKVRRHDMRVHPYQRIVTADRAATGN], Genscript, 1:2000), anti-Qki-6 (Sigma–Aldrich, AB9906, 1:2000), anti-αB-crystallin (Abcam, ab13496, 1:2500), anti-αA-crystallin (Abcam, ab5595, 1:5000), anti-β-crystallin (Santa Cruz Technology, sc-22745, 1:200), anti-β-actin (Sigma–Aldrich, A5441, 1:10000) overnight at 4 °C followed by incubation with the corresponding horse-radish peroxidase (HRP)-conjugated secondary antibodies and detection using a SuperSignal enhanced chemiluminescence system (Thermo Fisher Scientific). Immunoblotting images were quantified using Image Studio (LI-CoR). For co-IP, HLE-B3 cells and NLPCs were washed twice with the cold PBS and lysed in hypotonic lysis buffer (10 mM HEPES [pH 7.9], 1.5 mM MgCl$_2$, and 10 mM KCl, supplemented with freshly added protease inhibitors and DTT) for 10 min. The swollen cells were disrupted by a dounce homogenizer on ice followed by centrifugation at $13,000 \times g$ at 4 °C for 15 min. The pellet (nuclei) was resuspended in low-salt buffer (20 mM HEPES [pH 7.9], 1.5 mM MgCl$_2$, 20 mM KCl, 0.2 mM EDTA, and 25% glycerol, supplemented with freshly added protease inhibitors and DTT) followed by centrifugation at 13,000 g at 4 °C for 10 min. The pellet was fixed by 1% formaldehyde in NP-40 buffer (50 mM Tris-HCl [pH 7.4], 150 mM NaCl, 5 mM EDTA, and 0.05% NP-40, supplemented with freshly added protease inhibitors) for 10 min followed by quenching with 0.125 M glycine and washing with cold PBS. The crosslinked pellet was then sonicated using a Bioruptor Pico sonication device (Diagenode) for 60 cycles (30 s on, 30 s off) on high power setting. After centrifugation at $13,000 \times g$ at 4 °C for 10 min to remove insoluble debris, the supernatant (nuclear fraction) was incubated with antibodies against Qki-5 (immunizing rabbit with a short synthetic peptide [CGAVATKVRRHDMRVHPYQRIVTADRAATGN], Genscript), Qki-6 (Sigma–Aldrich, AB9906), Srebp2 (10007663, Cayman Chemical), or normal rabbit immunoglobulin G (CST, 2729) at 4 °C overnight followed by further incubation in the presence of magnetic recombinant protein G coated-beads (Thermo Fisher Scientific) for another 2 hr at 4 °C. Bound beads were then washed 3 times in cold NP-40 buffer by inverting the tubes, boiled in sample buffer at 95 °C for 20 min, and subjected to sodium dodecyl sulfate-polyacrylamide gel electrophoresis (SDS-PAGE) and immunoblotting with anti-QKI-5 (1:1000), anti-SREBP2 (1:500), anti-Flag (Sigma–Aldrich, 1804, 1:500), and anti-HA (Abcam, ab18181, 1:500).

**Subcellular fractionation.** HLE-B3 cells were cultured in lipid depletion condition described above, collected and washed in PBS, and resuspended in lysis buffer containing 10 mM HEPES [pH 7.4], 10 mM KCl, 0.05% NP-40, protease inhibitor cocktail, phosphatase inhibitor, and dithiothreitol). Protein lysates were incubated for 20 min on ice and centrifuged at $14,000 \times g$ at 4 °C for 10 min. The supernatants containing the cytoplasmic proteins were separated, and the cell pellets were washed, resuspended in low-salt buffer (10 mM Tris-HCl [pH 7.4], 0.2 mM MgCl$_2$, 1% Triton X-100 and incubated for 15 min on ice. After centrifugation at $14,000 \times g$ at 4 °C for 10 min, the supernatants containing nuclear soluble proteins were separated. The cell pellets were then resuspended in 0.2 N HCl and incubated for 20 min on ice followed by 10 min of centrifugation at $14,000 \times g$ at 4 °C. The Supernatant containing chromatin proteins were then neutralized with the equal volume of 1 M Tris-HCl [pH 8.0]. Each fraction was quantified and proceeded to

immunoblotting assay with the following primary antibodies: anti–QKI-5 (immunizing rabbit with a short synthetic peptide [CGA–VATKVRRHDMRVHPYQRIVTADRAATGN], Genscript, 1:1000), anti–Qki-6 (Sigma–Aldrich, AB9906, 1:500), anti-SREBP2 (Abcam, ab30682, 1:500), anti-GAPDH (Santa Cruz Biotechnology, SC-32233, 1:5000), anti-SP1 (Abcam, ab13370, 1:500), and anti-Histone H3 (Abcam, ab5176, 1:5000)[107].

**Cholesterol measurement.** The lens and whole brain tissues of control and Nestin-CreER$^{T2}$;Qk$^{L/L}$ mice at P19 were homogenized in the lipid extraction buffer (chloroform: isopropanol: NP-40 = 7: 11: 0.1) (200 μl of the lipid extraction buffer per 10 mg of tissue). After centrifugation at $13,000 \times g$ at 4 °C for 10 min, the supernatant was collected and air-dried to remove the organic solvent. Dried lipid was used to measure cholesterol levels by Total Cholesterol Assay Kits (Cell Biolabs) according to the manufacturer's instructions. For plasma cholesterol measurement, blood was collected retro-orbitally from control and Nestin-CreER$^{T2}$; Qk$^{L/L}$ mice at P19, immediately followed by treatment of 0.5 M EDTA as an anticoagulant. Then, the blood samples are centrifuged at $2000 \times g$ at 4 °C for 10 min, and the top layer was used with 1:50 dilution for cholesterol measurement as shown above.

**Lanosterol treatment.** Two microliters of lanosterol eye drop (Lanomax, Ventura Laboratories Brea California 92821 USA) was directly applied to the eyes of control and Nestin-CreER$^{T2}$;Qk$^{L/L}$ mice twice a day for 7 days starting from the age of P14. This study was approved by MD Anderson's animal facility. The dissected lenses at P21 were immediately imaged by Leica DFC450 C microscope camera, and aggregation index was quantified using Image J (NIH).

**Reverse transcriptase qPCR (RT-qPCR).** Total RNA was isolated using a RNeasy Mini Kit (Qiagen) as per the manufacturer's instructions. Two micrograms of RNA was reverse transcribed to cDNA by SuperScript III First-Strand Synthesis SuperMix (Thermo Fisher Scientific). Real-time RT-qPCR was performed using the iTaq Universal SYBR Green Supermix (Bio-Rad) with a 7500 Fast Real-Time PCR system (Applied Biosystems). The quantitative mRNA level of cholesterol biosynthesis genes was assessed according to the ΔΔCT method using Actb (mouse eye lens tissue) and GAPDH (NLPCs and HLE-B3 cells) as reference genes. List of oligonucleotide primer pairs used in RT-qPCR is provided in Supplementary Table 1.

**RNA-sequencing and pathway enrichment analysis.** Total RNA was isolated from freshly dissected lenses from control ($n = 3$) and Nestin-CreER$^{T2}$;Qk$^{L/L}$ ($n = 3$) mice at P17-19 using a RNeasy Mini Kit with the treatment of DNase (Qiagen). RNA sequencing was conducted by the Illumina HiSeq/MiSeq sequencing service at the MD Anderson Sequencing and Microarray Facility. The stranded paired-end RNA-sequencing analysis procedure was referenced to Pertea's protocol[108]. In brief, STAR (2.6.1b) program[109] was used to align pair-end RNA sequence data against the mouse reference genome (mm10) and the human reference genome (hg19) version (GRCh37.primary_assembly.genome) with STAR-2.6.1b with parameters–outSAMunmapped Within–outFilterType BySJout–twopassMode Basic–outSAMtype BAM SortedByCoordinate. After applying the HTSeq (0.11.0)[110] to extract the raw count tables based on the aligned bam files, DESeq2[111] was used to perform normalization and differential gene expression analysis. Differentially expressed genes ($p$-value < 0.05) were used for analysis of canonical cellular pathways and upstream transcription regulators altered upon Qki depletion by Ingenuity pathway analysis (IPA) (Qiagen Inc.).

**ChIP-seq and ChIP-qPCR.** The ChIP assays were performed as described previously[112]. In brief, NLPCs and HLE-B3 cells were crosslinked by 1% formaldehyde for 10 min and quenched by 0.125 M glycine. Then the crosslinked cells were suspended in ChIP lysis buffer (50 mM Tris-HCl [pH 7.4], 500 mM NaCl, 2 mM EDTA, 1% Triton X-100, 0.1% SDS, and 0.1% sodium deoxycholate, with freshly added protease inhibitors) and sonicated to 200–300 bp. The sheared chromatin was diluted in ChIP dilution buffer (50 mM Tris-HCl [pH 7.4], 100 mM NaCl, 2 mM EDTA, 1% Triton X-100, and 0.1% sodium deoxycholate, with freshly added protease inhibitors) at a ratio of 1:1, and then incubated with anti–Qki-5 (immunizing rabbit with a short synthetic peptide [CGA–VATKVRRHDMRVHPYQRIVTADRAATGN], Genscript), anti-Srebp2 (10007663, Cayman Chemical), anti-Pol II (Abcam, ab817), or normal rabbit IgG (CST, 2729) antibodies overnight at 4 °C. After incubation with prewashed magnetic recombinant protein G coated-beads (Thermo Fisher Scientific) for 2 hr, the protein-DNA complexes were washed three times with high-salt buffer (50 mM HEPES [pH 7.5], 500 mM NaCl, 1 mM EDTA, 1% Triton X-100, 0.1% sodium deoxycholate, and 0.1% SDS with freshly added protease inhibitors), twice with low-salt buffer (10 mM Tris-HCl [pH 8.0], 250 mM LiCl, 1 mM EDTA, 0.5% NP-40, and 0.5% sodium deoxycholate, with freshly added protease inhibitors), and once with TE buffer (10 mM Tris-HCl [pH 8.0] and 1 mM EDTA). Elution and reverse crosslinking were carried out in the elution buffer (50 mM Tris-HCl [pH 8.0], 10 mM EDTA, and 1% SDS) at 65 °C for 4 hr. After digestion with RNase A (Thermo Fisher Scientific) and proteinase K (Promega) for 1 hr at 55 °C, DNA samples were purified using a PCR purification kit (Qiagen). Library preparation was performed under the instructions of the KAPA Hyper Prep Kit (Kapa

Biosystems) and sequenced by Illumina HiSeqX Ten or NovaSeq 6000 (Jiangxi Haplox Clinical Lab Cen, Ltd). The FASTQ data were trimmed by Trim Galore (v0.4.4_dev) and mapped to the mouse genome (mm10 version) or human genome (hg19 version) using Bowtie (v1.2.2)[113], then peaks were identified by macs2 (v2.1.2)[114] with the parameters 'macs2 callpeak -f BAM -g mm/hs -q 0.05 -t ChIP. bam -n NAME -c INPUT.bam'Homer (v4.10.1)[115] with the following steps: makeTagDirectory and findPeaks Sample_tag -style factor -size auto -minDist default -i Input_tag -fdr 0.001. Signalplots and Heatmaps were generated with ngsplot[116]. ChIP-seq between WT and KO were normalized by total reads. IPA was used to analyze canonical cellular pathways enriched in genes whose promoters were co-occupied by Qki-5, Srebp2, and Pol II and in the overlapping gene cluster of Qki-5-bound genes in ChIP-seq and significantly downregulated genes in $Qk^{L/L}$ NLPCs and $QKI$ KO HLE-B3 cells relative to WT according to RNA-seq. Motifs enriched in QKI-5 peaks in NLPCs and HLE-B3 cells were identified by HOMER (v4.10.1) with the following parameters: findMotifsGenome.pl peaklist.bed mm10 –len given –len 6,8,10,12,14 –mis 2[115]. ChIP-qPCR was performed on a 7500 Fast Real-Time PCR system (Applied Biosystems) using iTaq Universal SYBR Green Supermix (Bio-Rad). List of oligonucleotide primer pairs used in ChIP-qPCR can be found in Supplementary Table 1.

**Protein expression and purification.** The full-length Qki-5, STAR domain, and KH3 domain of hnRNP K were cloned into an in-house modified version of pET-32a vector (Novagen, USA)[117]. A Thioredoxin (Trx)-His$_6$ tag and a PreScission protease cleavage site are present at the N-terminus of the multiple cloning sites. The resulting proteins contained a Trx-His$_6$ tag on the N-termini of full-length Qki-5, STAR domain, and KH3 domain of hnRNP K. The recombinant proteins were expressed in BL21 (DE3) Codon Plus *Escherichia coli* cells at 16 °C for 16–18 h. The cells were then lysed by an AH-1500 high pressure homogenizer (ATS Engineering Limited, China). The Trx-His$_6$-tagged protein complex was purified by Ni-NTA affinity chromatography (Qiagen, USA) followed by size-exclusion chromatography on a HiLoad 26/600 Superdex 200 (GE Healthcare, USA) in 50 mM Tris, pH 7.5, and 150 mM NaCl. Finally, protein was collected and con-centrated for MST and ITC experiments.

**Microscale thermophoresis.** MST experiments were performed on a Monolith NT.115 (NanoTemper, Germany) with a blue filter. Purified proteins were labeled by Att488-NHS, and the labeled proteins were separated from the free Atto488-NHS using a gravity column (PD MiniTrap G-25, GE Healthcare). 40 nM Atto488-labeled proteins (2X of the final reaction concentration) were prepared in PBS with 0.1% Tween-20, and oligonucleotides (QREs and QDEs) were dissolved and 2-fold serially diluted in PBS into 16 different concentration. Ten microliters of oligonucleotides of each con-centration was thoroughly mixed with 10 μl of 40 nM labeled proteins, and the mixture was incubated at room temperature for 10 min. Approximately 10 μl of the mixtures were then loaded into standard/premium treated capillaries. Measurement were per-formed at 25 °C using 60% MST power. Each experiment was repeated three times. Data analyses were performed using the NanoTemper analysis software.

**Isothermal titration calorimetry.** Isothermal titration calorimetry measurements were performed on a MicroCal$^{TM}$ iTC200 isothermal titration calorimeter (GE Healthcare, USA) in 50 mM Tris, [pH 7.5], and 150 mM NaCl. For ITC of QRE2 (UUCACUAACAA), 150 μM QRE2 was titrated into 14.5 μM Trx-His$_6$–Qki-5 and 14.3 μM Trx-His$_6$-STAR, respectively. For ITC of QDE1, 443 μM QDE1 was titrated into 42.6 μM Trx-His$_6$-Qki-5 and 44.1 μM Trx-His$_6$-STAR, respectively. For ITC of hnRNP K binding motif (CTCCCC), 150 μM CTCCCC was titrated into 15 μM Trx-His$_6$-KH3 of hnRNP K. The titration consisted of an initial injection of 0.4 μl followed by 19 injections of 2.0 μl every 120 s at 20 °C. The titration data and binding plot were analyzed with MicroCal Origin software with the one-site model.

**Statistical analysis.** Statistical analysis was performed using GraphPad Prism 8 soft-ware. The sample size was based on experimental feasibility, sample availability, and the number of necessary to obtain definitive results. The number of animals in each experiment is described in the corresponded Figure legends. Numerical results are presented as mean with error bars representing standard deviation (SD) except for Fig. 3f with standard error of the mean (SEM). For comparisons between the two groups, a two-tailed unpaired *t*-test was used except for Fig. 3f with a two-tailed paired *t*-test. Data distribution was assumed to be normal but has not been formally tested. All values of $p < 0.05$ were considered to be statistically significant. There were no rando-mization or blinding events during the experiments.

**Reporting summary.** Further information on research design is available in the Nature Research Reporting Summary linked to this article.

## Data availability
The RNA-and ChIP-seq data described herein have been deposited in the National Center for Biotechnology Information Gene Expression Omnibus and are accessible at GSE145475 and GSE144757. Source data are provided within this manuscript as a source data file. Source data are provided with this paper.

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

## Acknowledgements

We thank Liang Yuan and Chythra R. Chandregowda for mouse husbandry and care and all members of the Hu laboratory for insightful discussions. We thank Kenneth Dunner, Jr. for performing electron microscopy studies and Dr. Walter N. Hittelman for sharing confocal microscopy facility with us. We also thank Dr. Paul G. Leonard for sharing MST machine with us. We also thank Scientific Publications, Research Medical Library at MD Anderson for editorial assistance. This investigation was supported in part by grants from the Cancer Prevention & Research Institute of Texas (RP120348 and RP170002) and the National Cancer Institute (R37CA214800). J.H. is supported by The University of Texas Rising STARs Award, the Sidney Kimmel Scholar Award, the Sontag Foundation Distinguished Scientist Award, the Brockman Foundation, the Andrew Sabin Foundation, and MD Anderson Internal Research Grant. S.S. is supported by the Russell and Diana Hawkins Family Foundation Discovery Fellowship, Sam Taub and Beatrice Burton Endowed Fellowship in Vision Disease, and Roberta M. and Jean M. Worsham Endowed Fellowship. Y.C. is supported by R01GM130838 and BMS-MRA Young Investigator Award in Immunotherapy 569414. This research was supported by the University Cancer Foundation via the Institutional Research Grant program at the University of Texas MD Anderson Cancer Center.

## Author contributions

J.H. and S.S. designed the study and analyzed the data. S.S. conducted the experiments. S.S., H.Z., Q.M. and J.L. performed protein purification and binding assay. S.S., C.H. and F.L. performed ChIP and the bioinformatic analysis for ChIP-seq. S.S., Y.Wei, Y. Wang, and Y. C. performed RNA-seq and the bioinformatic analysis. T.S., X.Z., H.L. and Q.Z. helped preparing experimental model. A.Z. and S.W helped with mouse genotyping. J.H. and S.S. wrote the manuscript and prepared Figures.

## Competing interests

The authors declare no competing interests.
