## [Peer Review File · Nature Communications]

REVIEWER COMMENTS

Reviewer #1 (Remarks to the Author):

Qki activates ...

This is an important and impressive study that expands our understanding of the role(s) of cholesterol and its biosynthesis in cataracts. It represents a large body of work that is generally well performed.

Comments:

1. Disruption of Qki expression using a Nestin-Cre may not have been the best choice. This promoter clearly should have expression in other non-lens cells (especially in the nervous system). Does tamoxifen induction lead to systemic effects on cholesterol levels, etc.?
2. I am puzzled about how such limited expression can lead to such a dramatic effect. It appears that deletion is only achieved in about 10% of lens nuclei. How do the authors explain such a large opacity and such large biochemical changes? It would be nice to see some more detailed, higher resolution cellular analyses. Do changes in one cell affect the neighbors?
3. I find the immunostaining for ubiquitin and p62 problematic and potentially non-specific. I suggest that these images be deleted from the paper.
4. Alternatively, the microscopy and its analysis could be expanded. It might be very interesting to perform Z-series scans of the GFP staining and to correlate it with Qki deletion. The GFP staining seems to stain the entire cytoplasm of entire fiber cells. GFP staining seems to correspond to many more cells.
5. I would appreciate more discussion regarding how everything works. A modest deletion of a transcription factor that is only present/active in epithelial and superficial fiber cells leads to a severe cataract with aggregation of abundant fiber cell crystallins. Yet, this is totally reversed by lanosterol eye drops. Is the reversal permanent?
6. I am not convinced that the mass spec data add much. I am at all surprised that proteins as abundant as the crystallins were identified.

Reviewer #2 (Remarks to the Author):

The manuscript under review is focused on the regulation of cholesterol in the mammalian eye lens. Cholesterol has a tissue-selective role in the lens to help proper folding and presentation of the crystallin proteins that are essential for proper vision. Many studies show reduced cholesterol in the lens leads to cataracts likely due to misfolding on the crystallin protein network. Although the regulation of cholesterol balance is common to all cells, high capacity cells likely have unique aspects to the regulation and here the authors suggest that Qki, a STAR domain protein that is also expressed in the nervous system is involved in lens specific cholesterol regulation. Large deletions of the chromosomal region where Qki is located as well as more targeted deletions of Qki show its loss leads to cataracts. This paper focuses on the role of Qki in the regulation of cholesterol in the lens and by extension cataract formation. The studies nicely document that Qki localizes to chromatin and ChIP-seq studies show it colocalizes with SREBP-2 on promoters for cholesterol synthetic genes. Loss of SREBP-2 in the lens is also associated with cataract formation as well.

The protein-protein interaction studies suggest the two proteins interact together although these were performed on over-expressed transfected proteins. Also, mechanistically the authors provide data suggesting Qki binds to single stranded DNA at SREBP-2 target promoters but this mechanism is only superficially supported by the presented data.

Overall, the study provides a very nice start to a very interesting example of tissue specific regulation of cholesterol synthesis. However, there are critical issues that need to be addressed to support the authors model for how Qki works together with SREBP-2 to modulate gene expression. Key points to address include:

- 1). Protein-protein interaction studies were performed with over-expressed proteins, this approach can often times force interactions that do not occur in cells with endogenously expressed partners.
- 2). Authors claim the interaction and activity of Qki occurs in sterol depleted cells but there is no comparison with sterol replete cells for comparison.

- 3). The two bands highlighted as different forms of mature SREBP-2 in Fig. 6A are not likely both mature SREBP-2, the higher band that is present in cytosol and soluble nuclear extract is likely the full length precursor protein
- 4). The binding of Qki to SSDNA is interesting and the dependence on the TAA motif is apparent in the data in Fig. 7. However, the relevance to SREBP-2 binding and whether there are different motifs for both proteins is not established.
- 5). An essential experiment is to determine whether Qki works in concert with SREBP-2 in a co-transfection assay or similar in vivo experiment if possible. Can you separate Qki binding from SREBP binding?
- 6). Does Qki over expression in a non-lens cell stimulate SREBP-2 dependent cholesterol synthesis?
- 7). Methods and figure legends are very abbreviated, much more detail is needed for the reader to be able to follow precisely what is presented.

Reviewer #3 (Remarks to the Author):

This manuscript by Shin et al shows that Qki is required for cholesterol biosynthesis in the eye and that loss of Qki in lens epithelium results in cataracts development. The authors use RNA seq and ChIP-seq to show the involvement of Qki in regulation of cholesterol biosynthesis genes and the presence of Qki at promotor regions of these genes. The authors continued by using isothermal titration calorimetry and microscale thermophoresis to investigate Qki interaction with single-stranded DNA. These studies confirm the role for Qki in regulation of cholesterol biosynthesis and show a clear and novel role for Qki in cataracts development. These studies are interesting, especially together with the presented data of Qki interacting with single-stranded DNA interaction, however I feel that the Qki-DNA interaction findings are weak.

Major concerns:

- The majority of the data could be explained by an indirect role of Qki on cholesterol biosynthesis genes, namely via the regulation of SIRT2 expression and effect on PPAR and LXR (see point below on relevant literature). For example, Qki has been shown to regulated SIRT2 levels and SIRT2 promotes nuclear translocation of SREBP2. Or can be explained through interaction with SREBP2, which could be direct or indirect, indicated in the IP data in fig 4. The real evidence for DNA interaction is to my feeling weak, shown in ITC data of supplemental figure 6. The Qki interaction to DNA is much weaker than the binding to RNA. As I see it, the measurements of DNA interaction were in background range, while 3 times as much oligos and Qki were used in the assay, and the Kd uncertain to determine from the curve shown. ITC data should be repeated with the other QDEs as well as with a positive control (known DNA binding protein). The MST data shows clear effect by mutation in QDE, however due to the limit binding seen in the ITC assay, what is the 100% (ratio 1.0) of QDE in the MST assay? And please comment on the equal Kd of wild type QDE4 and mutated QDE5 seen from the MST assay?
- The authors indicate the TAA as binding sequence for Qki. Is this sequence enriched in genes found in Qki ChIP-seq data compared to unbound genes? Can this sequence be found in more promotor regions? Are there genes with an QDE and no SRE, and are these differently detected in the RNA seq of Qki WT vs KO and/or ChIP-seq of Qki? And can also a Qki binding half site be found?
- All western blots and immunostaining of multiple experiments should be quantified.
- The number of experiments preformed and time point should be more clarified in the figures or legends. Such as the n used for the qPCRs shown in fig 2d, 2j, 6i and 6j. And in fig 3b, where is stated number of counted cells = 10. Is this within 1 mouse, or 10 cells from tissue of each mouse and are more mice quantified? What are the squares in the graph indicating?
- The number of reads found in the Qki ChIP-seq is overall much lower than SREBP2 and PolIII.

This could be explained by Qki binding at many more genes than SREBP2, 'deluding'; the signal. However the scale of the Y-axis should at least be presented similar as the Y-axis of the input. And it would be good to also show IgG control ChIP-seq with similar Y-axis.

- For the transcriptomic profiling P17-19 is used (fig 2) "to exclude secondary effects". However, in fig 1 a clear phenotype is seen as P19. Please explain or correct.

- The authors only investigated Qki-5 isoform. While the KO model reduces all isoforms, as well as the knowledge that Qki forms heterodimers and also Qki-6 is found abundantly in the nucleus, other isoforms should be taken along in these studies, such as in immunostaining, western blots and IP of SERBP2. At least in figures 2e, 4a and 4e.

- Recent/relevant literature is not discussed, and should be checked and included. Such as:
1. QKI regulates adipose tissue metabolism by acting as a brake on thermogenesis and promoting obesity.

Lu H, Ye Z, Zhai Y, Wang L, Liu Y, Wang J, Zhang W, Luo W, Lu Z, Chen J.
EMBO Rep. 2020. PMID: 31868295

2. SIRT1 mediates the role of RNA-binding protein QKI 5 in the synthesis of triglycerides in non-alcoholic fatty liver disease mice via the PPAR α /FoxO1 signaling pathway.

Zhang W, Sun Y, Liu W, Dong J, Chen J.
Int J Mol Med. 2019. PMID: 30664220

3. Expression of Quaking RNA-Binding Protein in the Adult and Developing Mouse Retina.

Suiko T, Kobayashi K, Aono K, Kawashima T, Inoue K, Ku L, Feng Y, Koike C.
PLoS One. 2016. PMID: 27196066

4. Quaking promotes monocyte differentiation into pro-atherogenic macrophages by controlling pre-mRNA splicing and gene expression.

de Bruin RG, Shiue L, Prins J, de Boer HC, Singh A, Fagg WS, van Gils JM, Duijs JM, Katzman S, Kraaijeveld AO, Böhringer S, Leung WY, Kielbasa SM, Donahue JP, van der Zande PH, Sijbom R, van Alem CM, Bot I, van Kooten C, Jukema JW, Van Esch H, Rabelink TJ, Kazan H, Biessen EA, Ares M Jr, van Zonneveld AJ, van der Veer EP.

Nat Commun. 2016. PMID: 27029405

5. miR-29a promotes scavenger receptor A expression by targeting QKI (quaking) during monocyte-macrophage differentiation.

Wang S, Zan J, Wu M, Zhao W, Li Z, Pan Y, Sun Z, Zhu J.
Biochem Biophys Res Commun. 2015. PMID: 26056009

6. The QKI-PLP pathway controls SIRT2 abundance in CNS myelin.

Zhu H, Zhao L, Wang E, Dimova N, Liu G, Feng Y, Cambi F.
Glia. 2012. PMID: 21948283

Minor concerns:

- In fig 3f an un-paired, not paired, t-test should be used to determine p-value, this is not comparing repeated measurements.

- In figure 1 is shown that Qki-cKO results in accumulation of ubiquitin, while in figure 2h is shown that the protein ubiquitination pathway is the most downregulated pathway. Please include clarification/comment on this in the manuscript.

- The specific mice used as control animals can be made more clear. Are also mice used with partial Qki-cKO? Resulting in reduced levels, and does this result in changes in cholesterol biosynthesis and/or cataracts?

- In figures 2i-j only significant RNA-seq genes are validated by qPCR. Please also validate unchanged genes. Similar, in fig 6 for ChIP-seq data. And please also show genes where PolIII binding is unchanged or even increased upon Qki KO.

- Limit or missing information in the Methods section. Such as:

o For the ITC only the concentrations of QRE2 and QDE1 are indicated, was that similar for the

other QREs and QDEs?

- o What is the difference between QRE1 and QRE2?
- o Little detail on mass spec and electron microscopy.
- o No description for Qki-5 HA vector.

- Do the authors have insight on the regulation of Qki in lens epithelium? Is this changed by risk factors for cataracts?

- Please include Nestin staining in figure 1c/d.

Kind regards,
Janine van Gils

Rebuttal to reviewer comments for “*Qki activates Srebp2-mediated cholesterol biosynthesis for maintenance of eye lens transparency*”

We thank all three reviewers for their enthusiastic and constructive feedback. On the basis of the comments from the reviewers, we have performed 28 experiments and added 26 figure panels in the revised manuscript. In addition, we included 19 rebuttal figure panels in this point-to-point rebuttal letter for the purpose of clarification.

REVIEWER COMMENTS

Reviewer #1 (Remarks to the Author):

Qki activates ...

This is an important and impressive study that expands our understanding of the role(s) of cholesterol and its biosynthesis in cataracts. It represents a large body of work that is generally well performed.

We greatly appreciate the reviewer for recognizing that our study is “important and impressive” and “expands our understanding of the role(s) of cholesterol and its biosynthesis in cataracts”. We also appreciate the reviewer for praising our work was “well performed”.

Comments:

1. Disruption of Qki expression using a Nestin-Cre may not have been the best choice. This promoter clearly should have expression in other non-lens cells (especially in the nervous system).

We appreciate the reviewer for this important point regarding the lens-specific role of Qki in regulating cholesterol level, which greatly helped us describe our model more clearly in the revision. We agree with the reviewer that *Nestin-CreER^{T2}* is also expressed in non-lens cells including the neuronal lineage. In fact, our group previously showed that Nestin is expressed in the neural stem cells (NSCs) in the subventricular zone (SVZ) of the brain with the same mouse model (Shingu et al., 2016). However, although *Nestin-CreER^{T2}* is not exclusively expressed in the lens cell types, we have the following lines of evidence to show that lens cell-expressing Qki plays an essential role in regulating the lens cholesterol level in our mouse model. 1) Eye lens tissue is avascular, which limits entry of blood cholesterol and minimize the potential impact of cholesterol uptake originated from other tissues to the lens cells. 2) During the time frame used in this study (P7-P19), the NSC population is a very small fraction of the total neural cells in the brain and therefore it contributes little to the total cholesterol production in the brain. Supporting this notion, we showed that the cholesterol level in the whole brain was not altered in our mouse model at P19, the timepoint at which significant downregulation of cholesterol pathway in eye lens was observed (**Supplementary Fig. 4e, revised**). 3) Similarly, as we mentioned in the response to **Comment 1-1b** below, the plasma cholesterol level was not affected in our mouse model at P19 (**Supplementary Fig. 4f, revised**), suggesting that the cholesterol level reduction in the eye lens is mainly caused by Qki depletion in the eye lens cells. We also included this part in the revised text, line 217-219.

Of note, other Cre-alleles used in the eye lens studies from the literature also displayed expression of Cre recombinase in non-lens tissues. For example, Le-cre driven by Pax6 promoter is expressed in endocrine pancreas and other ectodermal eye structures, and α -crystallin promoter-driven MLR10-cre and MLR39-cre are expressed in midbrain, pituitary gland, and retinal cells (Chaffee et al., 2014; Lam et al., 2019; Scheiblin et al., 2014), suggesting the limitation of obtaining lens-specific Cre alleles. Nevertheless, Nestin-cre line has been used to study lens in other papers as Nestin is expressed in the lens cells as shown in **Supplementary Fig. 1b** and other studies (Calera et al., 2006; Cammas et al., 2012; Yang et al., 2000). In summary, on the basis of the above-mentioned evidence, we are confident that the cataract phenotype in our mouse model is majorly caused by the specific depletion of Qki in lens cells that leads to impairment of cholesterol production in the lens tissue.

Does tamoxifen induction lead to systemic effects on cholesterol levels, etc.?

We appreciate the reviewer for raising this important question. Anatomical studies in lens have shown that lens is far from the blood supply (Cenedella, 1996), which potentially restrains the systemic effect of lens tissue to alter the blood cholesterol level. In addition, by measuring the blood cholesterol level in *Qk-iCKO* mice compared to the control mice, we confirmed that no systemic alteration of cholesterol level was found upon Qki depletion at P19 as shown in **Supplementary Fig. 4f** of the revised manuscript.

2. I am puzzled about how such limited expression can lead to such a dramatic effect. It appears that deletion is only achieved in about 10% of lens nuclei. How do the authors explain such a large opacity and such large biochemical changes? It would be nice to see some more detailed, higher resolution cellular analyses.

We appreciate the reviewer for bringing up this critical point to help us better understand the cellular impact of Qki depletion at the tissue-level. In our revised manuscript, we have performed more in-depth characterization of efficiency of Qki depletion induced by *Nestin-CreER^{T2}* promoter. We observed that Qki-5 expression was almost entirely abolished (**Fig. 1a, b, revised**) at P19 at higher resolution (40X), whereas the original figure examined at P17 showing approximately 10% of the lens cells deleted with Qki-5. This indicates that major *Qk* deletion events actively occur between P17 and P19, and the variability in efficiency of *Qk* deletion exists among the animals upon tamoxifen injection. While we intended to use P17 to indicate the early stage of Qki depleting process, we agree with the reviewer that P19 is more representative and less confusing. In summary, our data suggested that P19 is timely correlated with the onset of the cataract formation (**Fig. 1d, e, revised**) and downregulation of cholesterol biosynthesis (**Fig. 2a-f, revised**). We also include this point in the revised text, line 127-129.

Do changes in one cell affect the neighbors?

Thank you for raising this excellent point. Based on the Qki-5 and GFP co-staining, Qki-5 depletion is greatly correlated with GFP expression, indicating the Cre-specific depletion of Qki-5 expression. We also observed that at P19, GFP is expressed in the superficial lens fiber cells, where nucleus is still intact (DAPI-positive), and the lens membrane structure is substantially impaired, as shown by immunostaining of lens-specific membrane protein called aquaporin 0 (AQP0) (**Fig. 1c and Supplementary Fig. 2a, revised**). Notably, lens fiber cells undergo

differentiation from the outer layer towards center of the lens, and one major process during differentiation is denucleation, which is important for maintaining transparency of the lens (Wride, 2011). Those terminally differentiated lens fiber cells without organelles are metabolically inactive, which requires nutrients/metabolites from the metabolically active neighboring fiber cells. The enclosed fiber cells are tightly packed and behave as a whole, similarly to a single cell in term of metabolic exchange (Mathias et al., 2007). Interestingly, we found that AQP0⁺GFP⁻ cells (terminally differentiated lens fiber cells that formed before Qki was depleted) were also impaired with regard to their cellular membrane structure (**Fig. 1c and Supplementary Fig. 2a, revised**), probably due to the indirect impact by Qki-depleted cells (GFP⁺ cells), although the impact of AQP0⁺GFP⁻ cells was not as remarkable as AQP0⁺GFP⁺ cells. On the basis of the data above, we believe that Qki-depleted cells could potentially affect the neighboring cells by reducing the metabolic supports such as cholesterol, and we have included this point of discussion in the revised manuscript. Please, see the revised text, line 430-442.

3. I find the immunostaining for ubiquitin and p62 problematic and potentially non-specific. I suggest that these images be deleted from the paper.

We appreciate this helpful suggestion from the reviewer. We agree that the immunostaining for ubiquitin and p62 might not be informative. Since we have shown the massive accumulation of p62, hsp90, and ubiquitin by tissue-level immunoblotting, which is sufficient to indicate the induction of protein aggregation upon Qki loss, we have deleted the immunostaining data in the revised manuscript.

4. Alternatively, the microscopy and its analysis could be expanded. It might be very interesting to perform Z-series scans of the GFP staining and to correlate it with Qki deletion. The GFP staining seems to stain the entire cytoplasm of entire fiber cells. GFP staining seems to correspond to many more cells.

Thank you for the great suggestion. With the reviewer's suggestion, we examined the Z-series of the lens tissue sections co-stained with anti-Qki-5 and anti-GFP antibodies using confocal microscopy and found that GFP expression is highly correlated with Qki-5 expression in control mice, and Qki-5 expression is nearly 100% depleted in the GFP⁺ cells (**Supplementary Fig. 1d, revised**), which is consistent with the data obtained from episcopic microscopy (**Fig. 1a, b, revised**). The GFP protein expressed by the mTmG reporter allele (Muzumdar et al., 2007) is a membrane-bound GFP protein, which could be confirmed by the co-staining with the lens-specific membrane protein AQP0 in control mice (**Fig. 1c, Supplementary Fig. 2a, revised**). Interestingly, with the high-resolution analysis of GFP localization, we found the lens membrane structure was largely affected in lens fiber cells with mislocalization of membranes indicated by the staining of GFP and AQP0. Importantly, the lens fiber cells with Qki depletion failed to elongate and migrate toward the center of the body and showed aberrant directionality. Lens cell membrane has a molar ratio of cholesterol/phospholipids that is 2-4-fold higher than those of other typical cell membranes, and the unusually abundant cholesterol in lens membrane is critical for α -crystallin binding to the lens membrane for its chaperone activity to prevent from abnormal crystallin aggregation (TANG et al., 1998). We have included this point in the revised text, line 127-141.

5. I would appreciate more discussion regarding how everything works. A modest deletion of a transcription factor that is only present/active in epithelial and superficial fiber cells leads to a severe cataract with aggregation of abundant fiber cell crystallins. Yet, this is totally reversed by lanosterol eye drops.

As in the response to **Comment 1-2** shown above, Qki depletion was nearly 100% at P19 in both lens epithelial cells and lens fiber cells when we start to observe cataract phenotype. We also showed that Qki depletion in differentiating lens fiber cells indirectly affect the integrity of the membrane structure and the entire cell morphology of terminally differentiated lens fiber cells, potentially due to the defect in the metabolic support as the enclosed bag of regularly ordered lens fiber cells form a micro-circulatory system to exchange fluid contents between neighboring cells (Mathias and Kistler, the lens circulation, 2007).

We postulate that this tissue-level effect of Qki depletion ultimately leads to the massive and fast progression of cataract phenotype. We included this point in the revised manuscript as a part of the discussion, line 427-442.

We started the lanosterol treatment at P14 before the cataract phenotype initiated to ask if the sterol repletion could help slow down/prevent from the cataract formation, and we observed the great reduction of cataract formation at P21, immediately after completion of one week-lanosterol treatment. Please, see another example of the *Qk-iCKO* lens in **(Rebuttal Fig. 1)** indicating the impact of lanosterol treatment.

Rebuttal Figure 1. Another set of representative images of lenses isolated from *Qk-iCKO* mice given either mock (control) or lanosterol-based treatment for one week. Scale bar: 0.5 mm.

Is the reversal permanent?

To address whether the impact of lanosterol treatment could be prolonged, and ultimately prevented eye lens from formation of cataract permanently, we analyzed the aggregation index of the eye lens isolated from Ctrl and *Qk-iCKO* mice 5 days after the completion of one week-lanosterol treatment. Alleviation of cataract phenotype in Qki-depleted lens induced by lanosterol was not as substantial as the impact we observed immediately after the treatment **(Rebuttal Fig. 2a-c)**.

Of note, there was a variation in the degree of rescuing cataract phenotype by lanosterol, suggesting that exogenous lanosterol effect could prolong to different extents in the eye lens among different animals after the treatment was stopped. Due to the limitation of monitoring the delivery efficiency and stability of lanosterol in the lens tissue, further investigation is required to understand if the constant lanosterol treatment can help prevent cataract formation. Yet, based on our data, we believe that the lanosterol treatment can slow down the cataract formation induced by Qki depletion, but the effect of the treatment is not permanent.

Rebuttal Figure 2. **a** Timeline and schematic of lanosterol-based treatment in mice. Analysis was performed at P26 **b** Representative images of lenses isolated from Ctrl and *Qk-iCKO* mice given either mock (control) or lanosterol-based treatment for 1 week (P14-P21) and analyzed at P26. Scale bar: 0.5 mm. **c** Quantification of the area of protein aggregates from isolated Ctrl and *Qk-iCKO* lenses after mock and lanosterol-based treatment. Aggregation index = $X / \text{average of the area of aggregates in Ctrl lenses with the mock treatment}$. X = area of aggregates. ns = not significant ($p \geq 0.05$; paired *t*-test). The results are presented as means with standard error of the mean (SEM) ($n = 3$).

6. I am not convinced that the mass spec data add much. I am at all surprised that proteins as abundant as the crystallins were identified.

We thank the reviewer for raising this point. We initially speculated that the big mass found in the eye lens of *Qk-iCKO* mice was abnormal growth of cell mass such as tumors because our lab had found that *Qki* is a tumor suppressor in the brain (Shingu et al., 2016). After we carefully validated that the isolated aggregates were insoluble to regular protein lysis buffers (e.g., RIPA buffer) and only soluble to 8M urea, we started to consider the possibility of *Qk-iCKO* mice as a cataract model. It might not be surprising to an expert who studies cataracts when she/he sees high enrichment of crystallin in these aggregates, but it was quite compelling for cancer researchers like us to exclude the possibility of abnormal cell growth in the eyes. Crystallin enrichment in the aggregates analyzed by mass spec allowed us to further confirm that our mouse model displayed pathogenesis of cataract development and focus on the mechanism by which *Qki* is required for lens transparency. We strongly agree with the reviewer that crystallin abundance is not a novel finding in our model, but rather a validation for the cataract model.

Reviewer #2 (Remarks to the Author):

The manuscript under review is focused on the regulation of cholesterol in the mammalian eye lens. Cholesterol has a tissue-selective role in the lens to help proper folding and presentation of the crystallin proteins that are essential for proper vision. Many studies show reduced cholesterol in the lens leads to cataracts likely due to misfolding on the crystallin protein network. Although the regulation of cholesterol balance is common to all cells, high capacity cells likely have unique aspects to the regulation and here the authors suggest that Qki, a STAR domain protein that is also expressed in the nervous system is involved in lens specific cholesterol regulation. Large deletions of the chromosomal region where Qki is located as well as more targeted deletions of Qki show its loss leads to cataracts. This paper focuses on the role of Qki in the regulation of cholesterol in the lens and by extension cataract formation. The studies nicely document that Qki localizes to chromatin and ChIP-seq studies show it colocalizes with SREBP-2 on promoters for cholesterol synthetic genes. Loss of SREBP-2 in the lens is also associated with cataract formation as well.

The protein-protein interaction studies suggest the two proteins interact together although these were performed on over-expressed transfected proteins. Also, mechanistically the authors provide data suggesting Qki binds to single stranded DNA at SREBP-2 target promoters but this mechanism is only superficially supported by the presented data.

Overall, the study provides a very nice start to a very interesting example of tissue specific regulation of cholesterol synthesis. However, there are critical issues that need to be addressed to support the authors model for how Qki works together with SREBP-2 to modulate gene expression.

We greatly appreciate the reviewer for considering that our study “provides a very nice start to a very interesting example of tissue specific regulation of cholesterol synthesis”, which has tremendously important physiological and pathological importance. We also appreciate reviewer’s insightful comments on our mechanism studies, which have helped us improve our manuscript significantly.

Key points to address include:

1). Protein-protein interaction studies were performed with over-expressed proteins, this approach can often times force interactions that do not occur in cells with endogenously expressed partners.

We thank the reviewer for this important question. We have performed the endogenous co-immunoprecipitation (Co-IP) for both Qki-5 and Srebp2 in HLE-B3 cells and found that endogenous protein-protein interaction was also present in the lens cells (**Supplementary Fig. 4h, I, revised**). In addition, we also showed the endogenous interaction between Srebp2 and Qki-5 by anti-Srebp2 Co-IP in the neural stem cells-derived lens progenitor-like cells (NLPCs) as well (**Fig. 4e**). We included this point in the revised text, line 245-249 and 259.

2). Authors claim the interaction and activity of Qki occurs in sterol depleted cells but there is no comparison with sterol replete cells for comparison.

We appreciate the reviewer for this excellent question. We hypothesized that if Qki transcriptionally cooperates with Srebp2, the chromatin-associated Qki isoform, Qki-5 can interact with the transcriptionally active form of nuclear Srebp2, which is predominantly translocated to the nucleus in response to sterol depletion condition. As suggested by the reviewer, we compared the interaction of Qki-5 and Srebp2 in sterol depletion condition to that in the sterol repletion condition (50 μ g/mg 25-hydroxycholesterol, 3 hr) from the nuclear lysate of HLE-B3 cells (Adams et al., 2004). We found that interaction between Qki-5 and Srebp2 was reduced in sterol repleted cells in IPs with both anti-Qki-5 and anti-Srebp2 antibodies (**Supplementary Fig. 4h, i, revised**), which suggests that interaction of Qki-5 with Srebp2 can be induced by sterol-low condition to meet the high demand of cholesterol biosynthesis in the lens cells. Interestingly, we observed the decreased protein levels of Srebp2 and Qki-5 in the nuclear lysate input (**Supplementary Fig. 4h, i, revised**). It is possible that decreased nuclear localization of Srebp2 and/or Qki-5 expression/stability are the other factors that contribute to the reduced binding between Qki-5 and Srebp2 in the lens cells. We included this point in the revised text, line 249-258.

3). The two bands highlighted as different forms of mature SREBP-2 in Fig. 6A are not likely both mature SREBP-2, the higher band that is present in cytosol and soluble nuclear extract is likely the full length precursor protein

We appreciate the reviewer for pointing out this issue, and we apologize for our oversight. We have revised the **Fig. 6a** by adding a molecular weight maker of 75 kD to indicate that these two bands are in the molecular size range of 55 - 65 kD, where mature SREBP2 is found. It was interesting that the lower band of Srebp2 blotting was predominantly found in the chromatin-bound fraction, which was more obvious when we performed subcellular fractionation compared to the total lysate. We found that a number of studies have presented two bands for mature SREBP2 (Kamisuki et al., 2009; Moon et al., 2019; Sakai et al., 1996; Seo et al., 2011; Suzuki et al., 2010), however the exact functions of these two bands are not clear. We believe that this is a very interesting phenomenon, which might indicate that there is additional processing of SREBP2 protein that hasn't be appreciated before. However, we hope that the reviewer would agree that it is out of the scope of the current study. Nevertheless, we found that these two bands are specific to SREBP2 as they were both decreased in SREBP2-depleted cells generated by the CRISPR-Cas9 system (**Supplementary Fig. 9a, revised**).

4). The binding of Qki to SSDNA is interesting and the dependence on the TAA motif is apparent in the data in Fig. 7. However, the relevance to SREBP-2 binding and whether there are different motifs for both proteins is not established.

Thank you for bringing up this important question, which inspired us to better understand co-dependency between Qki and SREBP2 with regard to chromatin localization. To address whether DNA binding of Qki is dependent on the presence of SREBP2 on the chromatin, we utilized the CRISPR-Cas9 system to delete SREBP2 in the HLE-B3 cells. Using the *SREBF2* KO line, which displayed the most efficient reduction of SREBP2 expression (**Supplementary Fig. 9a, revised**), we showed that the genome-wide chromatin localization of QKI-5 was not changed upon SREBP2 depletion (**Supplementary Fig. 9b, c, revised**), suggesting that SREBP2 is dispensable for QKI-5 binding to chromatin at the whole genomic level. Of note, those genes

that are co-targeted by QKI-5 and SREBP2 such as *MVD*, *CYP51*, and *FDPS*, are selectively reduced compared to the QKI-5 target genes that are not bound by SREBP2 (**Supplementary Fig. 6d**). Taken together, these data indicated that the recruitment of QKI-5 and SREBP2 to the promoters of cholesterol biosynthesis genes might be co-dependent. We believe that this is a very interesting question worthy of further investigation, and in fact, we are currently working on unraveling how Qki is recruited/regulated by other factors. We also included this point in the revised manuscript as a part of discussion, line 392-402.

In our study, we first reported the novel finding that Qki interacts with DNA via TAA motif. We chose the QKI DNA recognition elements (QDEs) containing TAA proximal to the defined sterol regulatory elements (SREs) on the SREBP2 target gene promoters. SRE is a relatively defined SREBP2 binding consensus sequence (typically containing CACCCCAC), although there are some levels of variations in sequences among different cholesterol synthesis genes (Sharpe and Brown, 2013). Therefore, Qki-5 binds to a distinct motif different from SRE. Yet, the proximity of SRE is important for Qki-5 binding as the QDEs that we tested in the binding assay are proximally downstream of the SREs of cholesterol biosynthesis genes discovered by previous studies (**Fig. 7b**). Please, see the revised text, line 378-382 in the revised manuscript for the clarification and discussion.

5). An essential experiment is to determine whether Qki works in concert with SREBP-2 in a co-transfection assay or similar in vivo experiment if possible.

We appreciate the reviewer for this important question. We examined whether the transcriptional activity of SREBP2 can be enhanced in concert with Qki-5 by using the sterol regulatory element (SRE)-luciferase reporter (pSynSRE-T-Luc) (Dooley et al., 1998). Our data showed that co-transfection of QKI-5 increased the transcriptional activity of SREBP2 (**Supplementary Fig. 7b, revised**), suggesting that QKI-5 functions as a coactivator of SREBP2. We also included this data in the revised text, line 354-357.

Can you separate Qki binding from SREBP binding?

As stated in **Comment 4**, we found that the QKI's binding to chromatin overall is not depend on the presence of SREBP binding genome-wide.

6). Does Qki over expression in a non-lens cell stimulate SREBP-2 dependent cholesterol synthesis?

Thank you for raising this excellent point. To ask the sufficiency of Qki in upregulation of SREBP2-dependent cholesterol synthesis, Doxycycline-inducible HA tagged QKI-5 was ectopically expressed in 293FT cells, and expression of cholesterol biosynthesis genes were examined. We found that SREBP2-mediated cholesterol biosynthesis genes were robustly upregulated including HMGCS1, HMGCR, IDI1, SQLE, LSS, CYP51, and MSMO1 (**Rebuttal Fig. 3a, b**). Notably, we also observed that nuclear form of SREBP2 was induced upon overexpression of QKI-5 (**Rebuttal Fig. 3b**). Collectively, we suggest that QKI is sufficient to trigger SREBP2-mediated cholesterol synthesis in non-lens cells.

Rebuttal Figure 3. a Results of RT-qPCR analysis of the genes involved in SREBP2-mediated cholesterol biosynthesis in HEK293FT with doxycycline-inducible ectopic expression of HA-tagged QKI-5 under sterol depletion condition. $n = 3/\text{group}$. * $p < 0.05$; ** $p < 0.01$; *** $p < 0.001$; **** $p < 0.0001$ (t -test). The results are presented as means with SD. **b** Immunoblots of mSREBP2, Qki-5–HA, and cholesterol biosynthesis enzymes, HMGCS1 and HMGCR. GAPDH: loading control. mSREBP2: mature SREBP2. $n = 3/\text{group}$.

7). Methods and figure legends are very abbreviated, much more detail is needed for the reader to be able to follow precisely what is presented.

Thank you for the comment. We have added more details in the methods and the figure legends.

Reviewer #3 (Remarks to the Author):

This manuscript by Shin et al shows that Qki is required for cholesterol biosynthesis in the eye and that loss of Qki in lens epithelium results in cataracts development. The authors use RNA seq and ChIP-seq to show the involvement of Qki in regulation of cholesterol biosynthesis genes and the presence of Qki at promotor regions of these genes. The authors continued by using isothermal titration calorimetry and microscale thermophoresis to investigate Qki interaction with single-stranded DNA. These studies confirm the role for Qki in regulation of cholesterol biosynthesis and show a clear and novel role for Qki in cataracts development. These studies are interesting, especially together with the presented data of Qki interacting with single-stranded DNA interaction, however I feel that the Qki-DNA interaction findings are weak.

We greatly appreciate the reviewer for recognizing our study on Qki in cataract development “clear and novel”. We also appreciate the reviewer for thinking our “studies are interesting, especially together with the presented data of Qki interacting with single-stranded DNA”, which clearly is a new finding that is going to change our understanding of Qki’s function. We would also like to thank the reviewer’s insightful comments on some of our mechanistic studies, which have helped us improve our manuscript greatly.

Major concerns:

- The majority of the data could be explained by an indirect role of Qki on cholesterol biosynthesis genes, namely via the regulation of SIRT2 expression and effect on PPAR and LXR (see point below on relevant literature). For example, Qki has been shown to regulated SIRT2 levels and SIRT2 promotes nuclear translocation of SREBP2. Or can be explained through interaction with SREBP2, which could be direct or indirect, indicated in the IP data in fig 4.

We appreciate the reviewer for raising this excellent point. We agree with the reviewer that SIRT2 might potentially play a role in Qki-Srebp2-dependent cholesterol biosynthesis as previous studies suggested the role of Qki in regulating Sirt-mediated signaling (Doetsch et al., 2002; Zhang et al., 2019; Zhu et al., 2012). To test this hypothesis, we first examined whether the expression level of Sirt family is altered upon Qki loss. In the transcriptomic profiles of the eye lens tissues, we found that Sirt2 is the most abundantly expressed one among Sirt family proteins (**Rebuttal Fig. 4a**). Yet, none of the Sirt family proteins was altered at the expression level in *Qk-iCKO* mice compared to control mice (**Rebuttal Fig. 4b**), suggesting that the expressions of Sirt family proteins are not dependent on Qki in the eye lens *in vivo*. In the RNA-seq data of the *in vitro* lens models such as NLPCs and HLE-B3 cells, we found that Sirt1, Sirt2, and Sirt7 are highly expressed (**Rebuttal Fig. 4c, d**), therefore the mRNA expressions of these proteins were further confirmed by RT-qPCR. In NLPCs, Sirt2 expression was reduced upon Qki loss in RNA-seq, which was also confirmed by RT-qPCR (**Rebuttal Fig. 4e, f**), whereas no alteration of SIRT2 expression was found upon QKI loss in HLE-B3 cells (**Rebuttal Fig. 4g, h**). The discrepancy between the *in vivo* and *in vitro* studies regarding the regulation of Sirt2 by Qki is not clear, however it has been widely accepted that the regulation of Sirt family protein is heavily influenced by the microenvironments, such as nutrients and metabolites in the culture media. More importantly, the direct role of Sirt2 in regulating SREBP2 has not been well established in the genetic study as the *Sirt2* KO mice did not display any changes in terms of the

expression levels of cholesterol biosynthesis genes and SREBP2 nuclear localization (Bobrowska et al., 2012). In addition, *Sirt2* KO mice did not show cataract phenotype, suggesting that Sirt2 is not a major determining factor of cataract formation (Bobrowska et al., 2012; McBurney et al., 2003). In summary, on the basis of the above-mentioned evidence, we believe that although Qki might regulate Sirt2 function/stability, the function of Sirt2 is not as critical as Qki in regulating SREBP2-mediated cholesterol and protein homeostasis in eye lens *in vivo*. On the basis of our data by taking advantage of Co-IP (**Fig. 4b-e**), ChIP-seq (**Fig. 4, Fig. 5, and Fig. 6**), DNA binding assay (**Fig. 7**), and transcription activity assay (**Supplementary Fig. 7b, revised**), our study indicates that Qki functions as a coactivator of SREBP2-mediated cholesterol biosynthesis, which represents one major mechanism to maintain the lens transparency.

Rebuttal Figure 4. a-b Relative read counts for Sirt family normalized to the read count of Sirt1 from RNA-seq data in (a) lenses from control mice (n = 3) in (b) lenses from control (n = 3) and *Qki*-iCKO (n = 3) mice. **c-d** Relative read counts for SIRT family normalized to the read count of SIRT1 from RNA-seq data in (c) WT NLPCs (n = 3) and in (b) WT HLE-B3 cells (n = 3). **e** Relative read counts for Sirt family normalized to the read count of Sirt1 from RNA-seq data in WT (n = 3) and *Qki*^{-/-} (n = 3) NLPCs. **f** Results of RT-qPCR analysis of *Sirt1*, *Sirt2*, and *Sirt7* in WT (n = 3) and *Qki*^{-/-} (n = 3) NLPCs. **g** Relative read counts for SIRT family normalized to the read count of SIRT1 from RNA-seq data in WT (n = 3) and *QKI* KO (n = 3) HLE-B3 cells. **h** Results of RT-qPCR analysis of *SIRT1*, *SIRT2*, and *SIRT7* in WT (n = 3) and *QKI* KO (n = 3) HLE-B3 cells. *p < 0.05; ****p < 0.0001; ns = not significant (p ≥ 0.05) (t-test). The results are presented as means with SD.

The real evidence for DNA interaction is to my feeling weak, shown in ITC data of supplemental figure 6. The Qki interaction to DNA is much weaker than the binding to RNA. As I see it, the measurements of DNA interaction were in background range, while 3 times as much oligos and Qki were used in the assay, and the Kd uncertain to determine from the curve shown. ITC data should be repeated with the other QDEs as well as with a positive control (known DNA binding protein).

We appreciate the reviewer for raising this important question. We agree with the reviewer that the ITC data shows much weaker interaction between Qki and DNA compared to the interaction between Qki and RNA. Under our experimental condition with the purified Qki-5 protein, we observed that the MST assay allowed us to have a better affinity of the Qki-5 protein to DNA compared to the ITC assay, and the DNA-Qki-5 binding was comparable to the RNA-Qki-5 binding in the MST assay (**Fig. 7a, d**). Difference regarding the thermodynamic properties of the purified Qki-5 protein in the MST and ITC assays is beyond our understanding. However, based on the literature, we can speculate that the sensitivity of the ITC assay is lower than that of the MST assay because ITC requires larger amount of proteins (μM-range) compared to MST (nM-range) (Seidel et al., 2013). Notably, Qki is known to form a homodimer via Qua1 domain (**Supplementary Fig. 8a, revised**) (Teplova et al., 2013). It is unclear whether the purified Qki protein can form a monomer or dimer under our binding condition, however the ITC assay may have less sensitivity to the monomer/dimer stoichiometry compared to the MST assay (Seidel et al., 2013).

We tested other QDEs in ITC assay as well, however, these sequences showed weak binding with Qki potentially due to our limitation of the experimental condition. To confirm our ITC data, we used another DNA binding protein, hnRNP K, as a positive control. hnRNP K behaves similarly to Qki as it also contains KH domains and binds to both DNA and RNA. Importantly, previous studies showed that hnRNP K binds to single-stranded nucleic acids in 1-3 μM-range, similar to the affinity of Qki in our MST assay. We purified the KH3 domain of hnRNP K (whose structure has been previously solved by X-ray crystallography), and tested for the interaction with the known binding motif CTCCCC in both MST and ITC assays (Backe et al., 2005). hnRNP K showed $K_d = 11.9 \mu\text{M}$ in the MST assay and $K_d = 13.8 \mu\text{M}$ in the ITC assay (**Rebuttal Fig. 5a, b**), which are both comparable to the binding between KH domain of Qki and QDE1 ($K_d = 11.9 \mu\text{M}$) in ITC assay.

Rebuttal Figure 5. a Titration curve for the MST assay showing the K_d value for fluorescently labeled Trx-His₆-hnRNP K and binding motif CTCCCC. **b** Result of ITC assay performed using Trx-His₆-hnRNP K and CTCCCC. Raw ITC data on 20 injections is shown at the top, and fitted curve is shown at the bottom.

The MST data shows clear effect by mutation in QDE, however due to the limit binding seen in the ITC assay, what is the 100% (ratio 1.0) of QDE in the MST assay?

Thank you for this important question, and we apologize for the confusion. In the MST assay, y-axis (fraction bound) indicates the amount of the Qki-5 protein bound to the 16 different concentrations (ranging from 7.6 nM to 250 μM) of the DNA as a ligand (x-axis). 100% in y-axis is 20 nM, the concentration of the Qki-5 protein used to test the binding affinity with DNA. Please, see the revised labeling for the MST plot in **Fig. 7d**.

And please comment on the equal K_d of wild type QDE4 and mutated QDE5 seen from the MST assay?

As the reviewer has pointed out, QDE4 ($K_d = 81 \mu\text{M}$) and QDE5-m ($K_d = 83 \mu\text{M}$) have very similar binding affinity. We observed a clear variation of Qki-binding affinities (K_d) among different TAA motif-containing sequences, ranging from the highest affinity of QDE1 with $2.01 \mu\text{M}$ to the lowest affinity of QDE4 with $81 \mu\text{M}$ under our experimental condition. We reasoned that although TAA is critical for the binding, the flanking sequences adjacent to TAA might also have negative or positive impact on the binding of Qki-5 with DNA. We haven't figured out the mechanism, but this is an important question that we are currently investigating with the systematic evolution of ligands by exponential enrichment (SELEX) or other high-throughput assays.

- The authors indicate the TAA as binding sequence for Qki. Is this sequence enriched in genes found in Qki CHIP-seq data compared to unbound genes? Can this sequence be found in more promoter regions? Are there genes with an QDE and no SRE, and are these differently detected in the RNA seq of Qki WT vs KO and/or CHIP-seq of Qki? And can also a Qki binding half site be found?

Thank you for raising this excellent point. Based on the Qki CHIP-seq data, we found that the frequency of TAA sequence in the Qk-bound promoters was similar to that in the total promoters. Besides, TAA is very prevalent (over 90% among all promoters) as the sequence is only three nucleotides (**Rebuttal Fig. 6a**). We also observed that TAA was slightly more prevalent in the promoter regions bound by Qki-5 together with SREBP2 compared to the entire Qki-5 bound promoter regions (**Rebuttal Fig. 6b**). Yet, the difference is very minimal, and we reasoned that Qki-5's interaction to the DNA is co-determined by other Qki-5-interacting proteins such as CTF/NF1, ATF4, NRF1, and SOX10,

as predicted by the motif analysis from the Qki-5 CHIP-seq data. As our binding assay was performed in a cell-free condition, where we were limited to test only the direct interactions between Qki-5 and DNA without the cellular environment, it is possible that the binding affinity is unable to recapitulate the interaction of Qki-5 to DNA in concert with other interacting

Rebuttal Figure 6. a Bar graph showing the TAA frequency (%) of QKI-5 bound promoters and total promoters in QKI-5 CHIP-seq data from HLE-B3 cells. **b** Bar graph showing the TAA frequency (%) of Qki-5 bound promoters and Qki-5 and Srebp2 co-bound promoters in Qki-5 and Srebp2 CHIP-seq data from NLPCs.

proteins as a whole complex. Since this is an essential point to further investigate, we have included a detailed discussion in the revised manuscript, line 451-470.

- All western blots and immunostaining of multiple experiments should be quantified.

We have quantified the western blots and immunostaining in the revised figures (**Fig. 1b**, **Supplementary Fig. 1f**, **Supplementary Fig. 3a**, and **Fig. 2f**).

- The number of experiments performed and time point should be more clarified in the figures or legends. Such as the n used for the qPCRs shown in fig 2d, 2j, 6i and 6j. And in fig 3b, where is stated number of counted cells = 10. Is this within 1 mouse, or 10 cells from tissue of each mouse and are more mice quantified? What are the squares in the graph indicating?

Thank you for this clarification. We modified the figure legends in our revised manuscript accordingly and added n for each qPCR data. Regarding the **Fig. 3b**, the 10 cells are from one mouse to show a representative data set, but in the **Fig. 3c**, we included 5 mice for each group to make our data statistically meaningful. The squares in the graph (Qk-iCKO group) just meant individual data points, and they were not different from the dots in the Ctrl group. To make the graph less confusing, we changed the squares to dots in the revision.

- The number of reads found in the Qki ChIP-seq is overall much lower than SREBP2 and PolIII. This could be explained by Qki binding at many more genes than SREBP2, 'deluding'; the signal. However the scale of the Y-axis should at least be presented similar as the Y-axis of the input. And it would be good to also show IgG control ChIP-seq with similar Y-axis.

Thank you for pointing out this issue, and we have corrected it in the revised manuscript. In addition, IgG control was compared to the ChIP-seq result in NLPCs. Please, see the revised manuscript with correction and addition (**Fig. 5e** and **Supplementary Fig. 5e, revised**). Besides, ChIP-qPCR validation of the most enriched cholesterol biosynthesis genes such as *Hmgcs1*, *Hmgcr*, and *Mvk* was performed with IgG control (**Rebuttal Fig. 7**).

Rebuttal Figure 7. Rabbit IgG (n = 3) and Qki-5 (n = 3) ChIP-qPCR analysis of the promoter regions of *Hmgcs1*, *Hmgcr*, and *Mvk* in NLPCs. ***p < 0.001; ****p < 0.0001 (t-test). The results are presented as means with SD.

- For the transcriptomic profiling P17-19 is used (fig 2) "to exclude secondary effects". However, in fig 1 a clear phenotype is seen as P19. Please explain or correct.

Thank you for pointing out this confusion for us to clarify. We corrected the statement as shown in the text, line 177-179. In our cataract model, one of the secondary effects we have observed was immune cell infiltration as we found that the Iba+ cells (potentially, macrophages) are highly infiltrated in the later timepoint (data not shown), which could mask the direct cause of cataract formation induced by Qki depletion. Collectively, we selected P17-19 as at this timepoint to profile pure lens cell population in the lens tissue for RNA-seq when cataract phenotype has initiated (**Fig. 1d, e, revised**).

- The authors only investigated Qki-5 isoform. While the KO model reduces all isoforms, as well as the knowledge that Qki forms heterodimers and also Qki-6 is found abundantly in the nucleus, other isoforms should be taken along in these studies, such as in immunostaining, western blots and IP of SERBP2. At least in figures 2e, 4a and 4e.

We appreciate the reviewer for this excellent point, which allows us to better understand the role of different Qki isoforms in eye lens. Previous studies have shown that different tissues express different Qki isoforms that display different subcellular localizations, as the reviewer clearly pointed out. For instance, Qki-6 is highly localized in the nucleus in the endothelial cells (De Bruin et al., 2016), while it shows predominate cytoplasmic localization in the oligodendrocytes of the brain (Hardy et al., 1996). Therefore, we further studied the biological role of Qki-6 in eye lens by examining the expression, subcellular localization, and the impact on the cooperation with SREBP2 in cholesterol biosynthesis. We found that Qki-6 is expressed in the eye lens cells (**Fig. 2e, Supplementary Fig. 1e, revised**) and particularly higher in the lens epithelial cells compared to the lens fiber cells (**Supplementary Fig. 1e, revised**). Besides, Qki-6 expression was more abundant in the nucleus compared to the cytosol (**Supplementary Fig. 1e and Supplementary Fig. 4j, revised**). Interestingly, while Qki-5 expression is nearly 100% depleted in the eye lens cells (both lens epithelial cell and lens fiber cells) from the Qk-iCKO mice at P19, the time point at which cholesterol biosynthesis is significantly downregulated and cataract is initiated (**Fig. 1a, b, revised**), Qki-6 still remained expressed (**Supplementary Fig. 1e, f, revised**). Particularly, at P19, Qki-6 expression was mostly intact in lens epithelial cells, whereas Qki-6 in the lens fiber cells was largely depleted (**Supplementary Fig. 1e, revised**). It is possible that Qki-6 is relatively more stable than Qki-5 and remained to be present at P19. We further showed that the interaction between Qki-6 and SREBP2 was very minimal (**Supplementary Fig. 4h, i, revised**). The data above provide us a clue that Qki-5 is the major Qki isoform in transcriptional regulation of cholesterol biosynthesis essential for the maintenance of the lens transparency.

- Recent/relevant literature is not discussed, and should be checked and included. Such as:

1. QKI regulates adipose tissue metabolism by acting as a brake on thermogenesis and promoting obesity.

Lu H, Ye Z, Zhai Y, Wang L, Liu Y, Wang J, Zhang W, Luo W, Lu Z, Chen J. EMBO Rep. 2020. PMID: 31868295

2. SIRT1 mediates the role of RNA-binding protein QKI 5 in the synthesis of triglycerides in non-alcoholic fatty liver disease mice via the PPAR α /FoxO1 signaling pathway.

Zhang W, Sun Y, Liu W, Dong J, Chen J.

Int J Mol Med. 2019. PMID: 30664220

3. *Expression of Quaking RNA-Binding Protein in the Adult and Developing Mouse Retina.*

Suiko T, Kobayashi K, Aono K, Kawashima T, Inoue K, Ku L, Feng Y, Koike C.

PLoS One. 2016. PMID: 27196066

4. *Quaking promotes monocyte differentiation into pro-atherogenic macrophages by controlling pre-mRNA splicing and gene expression.*

de Bruin RG, Shiue L, Prins J, de Boer HC, Singh A, Fagg WS, van Gils JM, Duijs JM, Katzman S, Kraaijeveld AO, Böhringer S, Leung WY, Kielbasa SM, Donahue JP, van der Zande PH, Sijbom R, van Alem CM, Bot I, van Kooten C, Jukema JW, Van Esch H, Rabelink TJ, Kazan H, Biessen EA, Ares M Jr, van Zonneveld AJ, van der Veer EP.

Nat Commun. 2016. PMID: 27029405

5. *miR-29a promotes scavenger receptor A expression by targeting QKI (quaking) during monocyte-macrophage differentiation.*

Wang S, Zan J, Wu M, Zhao W, Li Z, Pan Y, Sun Z, Zhu J.

Biochem Biophys Res Commun. 2015. PMID: 26056009

6. *The QKI-PLP pathway controls SIRT2 abundance in CNS myelin.*

Zhu H, Zhao L, Wang E, Dimova N, Liu G, Feng Y, Cambi F.

Glia. 2012. PMID: 21948283

We appreciate the reviewer for providing us this helpful resource. We have read all the following literature and discussed the papers, 1, 2, 4, and 6 in our revised manuscript and rebuttal letter.

Minor concerns:

- In fig 3f an un-paired, not paired, t-test should be used to determine p-value, this is not comparing repeated measurements.

We apologize for the confusion. We have used the two-tailed paired t-test here as we used the 6 pairs of the control and lanosterol treated eye lenses, and each pair is from one mouse, as one lens is used as a control and the other lens is used for the lanosterol treated condition.

- In figure 1 is shown that Qki-cKO results in accumulation of ubiquitin, while in figure 2h is shown that the protein ubiquitination pathway is the most downregulated pathway. Please include clarification/comment on this in the manuscript.

Thank you for raising this point. Those genes that are downregulated and clustered to the protein ubiquitination pathway upon Qki depletion are mainly involved in the degradation of ubiquitinated proteins, particularly ubiquitin-proteasome system (UPS). Most abundant genes that clustered in UPS belong to the major part of 26S proteasome subunits (23 genes) in the proteasomal degradation process. Notably, both 20S core particles such as PSMA2 and PSMB10 and 19S regulatory particles such as PSMC3 and PSMD11 are significantly decreased at the mRNA level (Livneh et al., 2016), suggesting the entire proteasomal degradation process is compromised, leading to the accumulation of undegraded poly-ubiquitinated proteins.

Deubiquitinating enzymes (DUBs) including UCHL1 and USP10 (Todi and Paulson, 2011) are also

downregulated, which can potentially lead to imbalance of ubiquitination, leading to abnormal ubiquitin accumulation. Of note, the impairment of ubiquitination cascade process was changed minorly, suggesting that the degradation is the major process impaired in Qki loss. We have included this point in the discussion section of the revised manuscript. Please, see the text, line 483-489 for the discussion.

- The specific mice used as control animals can be made more clear. Are also mice used with partial Qki-cKO? Resulting in reduced levels, and does this result in changes in cholesterol biosynthesis and/or cataracts?

Thank you for raising this question. Please, see the text, line 146-147 for the clarification of the control mice (including *Nestin-CreER^{T2}; Qk^{+/+}*, *Nestin-CreER^{T2}; Qk^{L/+}*, and *Qk^{L/L}*) we used in this study. We observed that *Nestin-CreER^{T2}; Qk^{L/+}* did not show cataract phenotype, suggesting the haploinsufficiency of Qki loss regarding cholesterol regulation, so we included this hetero model in our control group.

- In figures 2i-j only significant RNA-seq genes are validated by qPCR. Please also validate unchanged genes. Similar, in fig 6 for ChIP-seq data. And please also show genes where PolII binding is unchanged or even increased upon Qki KO.

Thank you for raising this question. Please, see the **Rebuttal Fig. 4e-h** as we have performed the RT-qPCR for the unchanged genes such as *Sirt7* in the NLPCs, and *SIRT1* and *SIRT2* in the HLE-B3 cells. Also, please see the representative genes that are not changed in Pol II binding on the promoter regions from ChIP-seq data (**Supplementary Fig. 7a, revised**).

- Limit or missing information in the Methods section. Such as:

o For the ITC only the concentrations of QRE2 and QDE1 are indicated, was that similar for the other QREs and QDEs?

For ITC, binding affinity of QDE2, QDE3, QDE4, QDE5 to Qki-5 was weak, so we did not include in the manuscript. Nevertheless, for the MST, we have included the concentrations for QDE2 (8.34 μ M), QDE3 (22.99 μ M), QDE4 (81 μ M), and QDE5 (38 μ M) shown in **Fig. 7c**.

o What is the difference between QRE1 and QRE2?

Thank you for this point. It was our oversight. QRE1 is CUUCUAAUAUAACUGCCUAAACUUUAAU (Galarneau and Richard, 2005), and QRE2 is UUCACUAACAA (Teplova et al., 2013) (underline indicates the motif). We have added this information in the revised manuscript.

o Little detail on mass spec and electron microscopy.

Thank you for your comment. We added more details on both experiments. Please, see the method in the revised manuscript.

o No description for Qki-5 HA vector.

Please, see the method in text, line 640-644 of the revised manuscript.

- Do the authors have insight on the regulation of Qki in lens epithelium? Is this changed by risk factors for cataracts?

We are glad that the reviewer pointed out this important aspect of the study. It is still unclear how expression and stability of Qki are regulated in eye lens as our study is the first to report the critical role of Qki in eye lens maintenance using the genetic mouse model. We have observed that Qki-5 is less stable compared to Qki-6. It will be interesting to further investigate what determines stability of different isoforms. Interestingly, we also observed that Qki-5 and Qki-6 were elevated upon cholesterol depletion at the protein level (**Supplementary Fig. 4h, i, revised**). Further study is needed to better understand how cholesterol availability can impact on the regulation of Qki in the eye lens.

The etiology of cataracts is associated with many risk factors including aging and oxidative stress. It is still remained unclear how Qki is altered under these risk factors. It is possible that the expression level of Qki is not changed, but the Qki activity can be potentially changed, which we are currently investigating. Interestingly, Doxorubicin, a type of chemotherapeutic agent, which is linked with cataracts by increasing the oxidative damage is shown to decrease the Qki level (Bayer et al., 2005; Gupta et al., 2018). This suggests that oxidative stress potentially can downregulate Qki expression, leading to cataracts.

- Please include Nestin staining in figure 1c/d.

Thank you for the comment. As another reviewer suggested us to remove this immunostaining data, we did not include this in our revised manuscript. Please, see the **Comment 1-3**.

*Kind regards,
Janine van Gils*

References

- Adams, C.M., Reitz, J., De Brabander, J.K., Feramisco, J.D., Li, L., Brown, M.S., and Goldstein, J.L. (2004). Cholesterol and 25-hydroxycholesterol inhibit activation of SREBPs by different mechanisms, both involving SCAP and Insigs. *Journal of Biological Chemistry* *279*, 52772-52780.
- Backe, P.H., Messias, A.C., Ravelli, R.B., Sattler, M., and Cusack, S. (2005). X-ray crystallographic and NMR studies of the third KH domain of hnRNP K in complex with single-stranded nucleic acids. *Structure* *13*, 1055-1067.
- Bayer, A., Evereklioglu, C., Demirkaya, E., Altun, S., Karslioglu, Y., and Sobaci, G. (2005). Doxorubicin-induced cataract formation in rats and the inhibitory effects of hazelnut, a natural antioxidant: a histopathological study. *Medical science monitor* *11*, BR300-BR304.
- Bobrowska, A., Donmez, G., Weiss, A., Guarente, L., and Bates, G. (2012). SIRT2 ablation has no effect on tubulin acetylation in brain, cholesterol biosynthesis or the progression of Huntington's disease phenotypes in vivo. *PLoS one* *7*, e34805.
- Calera, M.R., Topley, H.L., Liao, Y., Duling, B.R., Paul, D.L., and Goodenough, D.A. (2006). Connexin43 is required for production of the aqueous humor in the murine eye. *Journal of cell science* *119*, 4510-4519.
- Cammas, L., Wolfe, J., Choi, S.-Y., Dedhar, S., and Beggs, H.E. (2012). Integrin-linked kinase deletion in the developing lens leads to capsule rupture, impaired fiber migration and non-apoptotic epithelial cell death. *Investigative ophthalmology & visual science* *53*, 3067-3081.
- Cenedella, R.J. (1996). Cholesterol and cataracts. *Survey of ophthalmology* *40*, 320-337.
- Chaffee, B.R., Shang, F., Chang, M.-L., Clement, T.M., Eddy, E.M., Wagner, B.D., Nakahara, M., Nagata, S., Robinson, M.L., and Taylor, A. (2014). Nuclear removal during terminal lens fiber cell differentiation requires CDK1 activity: appropriating mitosis-related nuclear disassembly. *Development* *141*, 3388-3398.
- De Bruin, R.G., Van Der Veer, E.P., Prins, J., Lee, D.H., Dane, M.J., Zhang, H., Roeten, M.K., Bijkerk, R., De Boer, H.C., and Rabelink, T.J. (2016). The RNA-binding protein quaking maintains endothelial barrier function and affects VE-cadherin and β -catenin protein expression. *Scientific reports* *6*, 1-11.
- Doetsch, F., Petreanu, L., Caille, I., Garcia-Verdugo, J.-M., and Alvarez-Buylla, A. (2002). EGF converts transit-amplifying neurogenic precursors in the adult brain into multipotent stem cells. *Neuron* *36*, 1021-1034.
- Dooley, K.A., Millinder, S., and Osborne, T.F. (1998). Sterol regulation of 3-hydroxy-3-methylglutaryl-coenzyme A synthase gene through a direct interaction between sterol regulatory element binding protein and the trimeric CCAAT-binding factor/nuclear factor Y. *Journal of Biological Chemistry* *273*, 1349-1356.
- Galarneau, A., and Richard, S. (2005). Target RNA motif and target mRNAs of the Quaking STAR protein. *Nature structural & molecular biology* *12*, 691-698.
- Gupta, S.K., Garg, A., Bär, C., Chatterjee, S., Foinquinos, A., Milting, H., Streckfuß-Bömeke, K., Fiedler, J., and Thum, T. (2018). Quaking inhibits doxorubicin-mediated cardiotoxicity through regulation of cardiac circular RNA expression. *Circulation research* *122*, 246-254.
- Hardy, R.J., Loushin, C.L., Friedrich Jr, V.L., Chen, Q., Ebersole, T.A., Lazzarini, R.A., and Artzt, K. (1996). Neural cell type-specific expression of QKI proteins is altered in quakingviable mutant mice. *Journal of Neuroscience* *16*, 7941-7949.

Kamisuki, S., Mao, Q., Abu-Elheiga, L., Gu, Z., Kugimiya, A., Kwon, Y., Shinohara, T., Kawazoe, Y., Sato, S.-i., and Asakura, K. (2009). A small molecule that blocks fat synthesis by inhibiting the activation of SREBP. *Chemistry & biology* *16*, 882-892.

Lam, P.T., Padula, S.L., Hoang, T.V., Poth, J.E., Liu, L., Liang, C., LeFever, A.S., Wallace, L.M., Ashery-Padan, R., and Riggs, P.K. (2019). Considerations for the use of Cre recombinase for conditional gene deletion in the mouse lens. *Human genomics* *13*, 10.

Livneh, I., Cohen-Kaplan, V., Cohen-Rosenzweig, C., Avni, N., and Ciechanover, A. (2016). The life cycle of the 26S proteasome: from birth, through regulation and function, and onto its death. *Cell research* *26*, 869-885.

Mathias, R.T., Kistler, J., and Donaldson, P. (2007). The lens circulation. *Journal of Membrane Biology* *216*, 1-16.

McBurney, M.W., Yang, X., Jardine, K., Hixon, M., Boekelheide, K., Webb, J.R., Lansdorp, P.M., and Lemieux, M. (2003). The mammalian SIR2 α protein has a role in embryogenesis and gametogenesis. *Molecular and cellular biology* *23*, 38-54.

Moon, S.-H., Huang, C.-H., Houlihan, S.L., Regunath, K., Freed-Pastor, W.A., Morris IV, J.P., Tschaharganeh, D.F., Kasthuber, E.R., Barsotti, A.M., and Culp-Hill, R. (2019). p53 represses the mevalonate pathway to mediate tumor suppression. *Cell* *176*, 564-580. e519.

Muzumdar, M.D., Tasic, B., Miyamichi, K., Li, L., and Luo, L. (2007). A global double-fluorescent Cre reporter mouse. *genesis* *45*, 593-605.

Sakai, J., Duncan, E.A., Rawson, R.B., Hua, X., Brown, M.S., and Goldstein, J.L. (1996). Sterol-regulated release of SREBP-2 from cell membranes requires two sequential cleavages, one within a transmembrane segment. *Cell* *85*, 1037-1046.

Scheiblin, D.A., Gao, J., Caplan, J.L., Simirskii, V.N., Czymmek, K.J., Mathias, R.T., and Duncan, M.K. (2014). Beta-1 integrin is important for the structural maintenance and homeostasis of differentiating fiber cells. *The international journal of biochemistry & cell biology* *50*, 132-145.

Seidel, S.A., Dijkman, P.M., Lea, W.A., van den Bogaart, G., Jerabek-Willemsen, M., Lazic, A., Joseph, J.S., Srinivasan, P., Baaske, P., and Simeonov, A. (2013). Microscale thermophoresis quantifies biomolecular interactions under previously challenging conditions. *Methods* *59*, 301-315.

Seo, Y.-K., Jeon, T.-I., Chong, H.K., Biesinger, J., Xie, X., and Osborne, T.F. (2011). Genome-wide localization of SREBP-2 in hepatic chromatin predicts a role in autophagy. *Cell metabolism* *13*, 367-375.

Sharpe, L.J., and Brown, A.J. (2013). Controlling cholesterol synthesis beyond 3-hydroxy-3-methylglutaryl-CoA reductase (HMGCR). *Journal of Biological Chemistry* *288*, 18707-18715.

Shingu, T., Ho, A.L., Yuan, L., Zhou, X., Dai, C., Zheng, S., Wang, Q., Zhong, Y., Chang, Q., and Horner, J.W. (2016). Qki deficiency maintains stemness of glioma stem cells in suboptimal environment by downregulating endolysosomal degradation. *Nature Genetics*.

Suzuki, R., Lee, K., Jing, E., Biddinger, S.B., McDonald, J.G., Montine, T.J., Craft, S., and Kahn, C.R. (2010). Diabetes and insulin in regulation of brain cholesterol metabolism. *Cell metabolism* *12*, 567-579.

TANG, D., BORCHMAN, D., YAPPERT, M.C., and CENEDELLA, R.J. (1998). Influence of cholesterol on the interaction of α -crystallin with phospholipids. *Experimental eye research* *66*, 559-567.

Teplova, M., Hafner, M., Teplov, D., Essig, K., Tuschl, T., and Patel, D.J. (2013). Structure–function studies of STAR family Quaking proteins bound to their in vivo RNA target sites. *Genes & development* 27, 928-940.

Todi, S.V., and Paulson, H.L. (2011). Balancing act: deubiquitinating enzymes in the nervous system. *Trends in neurosciences* 34, 370-382.

Wride, M.A. (2011). Lens fibre cell differentiation and organelle loss: many paths lead to clarity. *Philosophical Transactions of the Royal Society B: Biological Sciences* 366, 1219-1233.

Yang, J., Bian, W., Gao, X., Chen, L., and Jing, N. (2000). Nestin expression during mouse eye and lens development. *Mechanisms of development* 94, 287-291.

Zhang, W., Sun, Y., Liu, W., Dong, J., and Chen, J. (2019). SIRT1 mediates the role of RNA-binding protein QKI 5 in the synthesis of triglycerides in non-alcoholic fatty liver disease mice via the PPAR α /FoxO1 signaling pathway. *International journal of molecular medicine* 43, 1271-1280.

Zhu, H., Zhao, L., Wang, E., Dimova, N., Liu, G., Feng, Y., and Cambi, F. (2012). The QKI-PLP pathway controls SIRT2 abundance in CNS myelin. *Glia* 60, 69-82.

REVIEWERS' COMMENTS

Reviewer #1 (Remarks to the Author):

My criticisms have been adequately addressed

Reviewer #2 (Remarks to the Author):

Revised paper is vastly improved and huge amount of new work included supports the overall theme discussed by the authors

Reviewer #3 (Remarks to the Author):

I would like to thank the authors for their extensive response on the concerns I raised. I'm happy with the additional data, information and comments. Although I do realize the manuscript is already dense with figures, I would suggest to add Rebuttal figure 5 to the supplemental data of the manuscript.

REVIEWERS' COMMENTS

Reviewer #1 (Remarks to the Author):

My criticisms have been adequately addressed

Reviewer #2 (Remarks to the Author):

Revised paper is vastly improved and huge amount of new work included supports the overall theme discussed by the authors

Reviewer #3 (Remarks to the Author):

I would like to thank the authors for their extensive response on the concerns I raised. I'm happy with the additional data, information and comments. Although I do realize the manuscript is already dense with figures, I would suggest to add Rebuttal figure 5 to the supplemental data of the manuscript.

We greatly appreciate the reviewer's insightful suggestions, which helped us to improve our manuscript in addressing the biochemical property of Qki and also Qki/Srebp2-mediated cholesterol biosynthesis at the transcriptional level. We have included the Rebuttal Figure 5 in the Supplementary Figure 9a, 9b.